# USING THE TRAINING HISTORY TO DETECT AND PREVENT OVERFITTING IN DEEP LEARNING MODELS

## ABSTRACT

Overfitting occurs in deep learning models when instead of learning from the training data, they tend to memorize it, resulting in poor generalizability. Overfitting can be (1) prevented (e.g., using dropout or early stopping) or (2) detected in a trained model (e.g., using correlation-based methods). We propose a method that can both detect and prevent overfitting based on the training history (i.e., validation losses). Our method first trains a time series classifier on training histories of overfit models. This classifier is then used to detect if a trained model is overfit. In addition, our trained classifier can be used to prevent overfitting by identifying the optimal point to stop a model's training. We evaluate our method on its ability to identify and prevent overfitting in real-world samples (collected from papers published in the last 5 years at top AI venues). We compare our method against correlation-based detection methods and the most commonly used prevention method (i.e., early stopping). Our method achieves an F1 score of 0.91 which is at least 5% higher than the current best-performing non-intrusive overfitting detection method. In addition, our method can find the optimal epoch and avoid overfitting at least 32% earlier than early stopping and achieve at least the same rate (often better) of achieving the optimal epoch as early stopping.

## 1 INTRODUCTION

Overfitting is one of the fundamental issues that plagues the field of machine learning (Nowlan & Hinton, 1992; Ng, 1997; Caruana et al., 2000; Cawley & Talbot, 2007; Erhan et al., 2010; Srivastava et al., 2014; Zhao et al., 2020), which can also occur when training a deep learning (DL) model. An overfit model increases the risk of inaccurate predictions, misleading feature importance, and wasted resources (Hawkins, 2004). Figure 1 shows example training histories (i.e., the training and validation losses curves) of an overfit and a non-overfit model. The training and validation losses of the overfit model both decrease at the beginning of the training process. Following that, the validation loss increases while the training loss decreases, resulting in a large gap between the training and validation losses. Such a trend indicates that the trained model is not generalizing well to new data.

Currently, the problem of overfitting is addressed by either (1) preventing it from happening in the first place or (2) detecting it in a trained model. Overfitting prevention methods stop overfitting from happening through methods such as early stopping (Morgan & Bourlard, 1989), data augmentation (Shorten & Khoshgoftaar, 2019), regularization (Kukačka et al., 2017), modifying the model by adding dropout layers (Srivastava et al., 2014) or batch normalization (Ioffe & Szegedy, 2015). Many of these methods are intrusive and require modifying the data or the model structure and expertise to execute correctly. Furthermore, even the non-intrusive prevention methods such as early stopping incur a trade-off between model accuracy and training time (Prechelt, 2012). For example, when using the early stopping method, stopping too late may

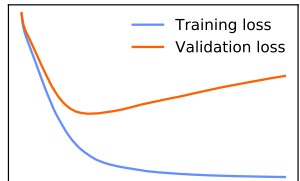

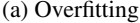

(a) Overfitting

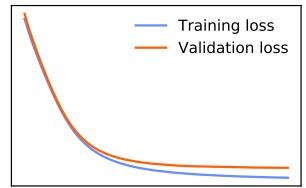

(b) Non-overfitting

Figure 1: Example training histories of overfit and non-overfit models.

improve model accuracy but also increase training time while stopping too early could result in a model that performs sub-optimally.

Overfitting detection methods typically attempt to identify if a trained model is overfit by retraining the model with noisy data points and observing the impact of these noisy data points on the model's accuracy (as an overfit model can learn the noise to reduce the impact) (Zhang et al., 2019). Alternatively, some detection methods check the hypothesis that the trained model and the data are independent, e.g., Werpachowski et al. (2019) check the hypothesis by comparing the test error with the estimated test error based on adversarial examples of the test set. However, similar to intrusive overfitting prevention methods, significant expertise is typically required to use existing detection methods. In addition, these methods require extra computational resources for activities such as generating adversarial examples, retraining the models, and converting the model.

In this paper, we are the first to propose a method for both overfitting detection and prevention based on training histories. Training histories have been used by researchers before to make decisions such as quantitative data acquisition and model selection (van Rijn et al., 2015; Strang et al., 2018; Bornschein et al., 2020; Mohr & van Rijn, 2021; 2022; Brazdil et al., 2022). Similarly, our method trains a time series classifier on a simulated dataset of training histories (i.e., labelled validation loss curves over epochs of training) of models that overfit the training data. Our trained time series classifier detects if a trained DL model is overfitting the training data by inspecting the validation loss history (which is captured as part of the training history). In contrast to existing overfitting detection methods, our method does not incur additional resources or costs since the training history is a byproduct of the training process. Additionally, our method (i.e., the trained time series classifier) can be used to prevent overfitting based on the validation losses of recent epochs (e.g., the last 20 epochs).

While we train our method on a simulated dataset, we evaluate it on a real-world dataset, collected from papers from top AI venues from the last 5 years. We collected the training histories from these papers that are explicitly labelled as overfitting or non-overfitting by the authors as the ground truth. Our results show that our method outperforms the state-of-the-art by at least 5% in terms of F-score for overfitting detection, with an F-score of 0.91. In addition, our method can prevent overfitting from happening at least 32% earlier than early stopping while having the same (and often better) rate of achieving the optimal epoch.

## 2 BACKGROUND AND RELATED WORK

### 2.1 OVERFITTING

Overfitting is a well-known and explored problem in the area of machine learning (Nowlan & Hinton, 1992; Ng, 1997; Caruana et al., 2000; Cawley & Talbot, 2007; Erhan et al., 2010; Srivastava et al., 2014; Zhao et al., 2020). Recent research has further noted the widespread presence and impact of overfitting in the sub-fields of machine learning including reinforcement learning (Song et al., 2020), adversarial learning (Rice et al., 2020), and recommender systems (Peng et al., 2021). For recommender systems that deal with massive amounts of data every day, incremental model updates are required to catch the most recent trend. However, the incrementally updated model may overfit to the most current data and forget previously learned knowledge (Peng et al., 2021). Song et al. (2020) study the observational overfitting regime in reinforcement learning, which overfits to only a small proportion of the observation space. Furthermore, Rice et al. (2020) report that overfitting happens more frequently in adversarial training than in traditional DL. Overfitting hurts the generalizability of a trained model, but generally predicting whether a model will overfit to a certain dataset before training it is formally undecidable (Bashir et al., 2020). In this paper, we study how to detect if a trained model is overfit and how overfitting can be prevented from happening during the training process. Below, we give an overview of existing methods to detect and prevent overfitting, and we describe the methods that we used as baselines to evaluate the accuracy of our method.

### 2.2 OVERFITTING DETECTION

In the field of symbolic regression, Kronberger et al. (2011) propose computing Spearman's non-parametric rank correlation coefficient (Spearman, 1987) between training and validation fitness (i.e., an evaluation metric for the symbolic regression model) to detect overfitting. Researchers

have also studied how to detect overfitting by injecting noise into the training data or generating new data. They typically retrain the model that is being tested with this noisy or new data and observe the impact on its performance to detect if the model is overfitting. For instance, Zhang et al. (2019) propose a Perturbation Validation (PV) method, which retrains the model after injecting different levels of noise (perturbation) into the labels. They retrain the model for each noise level and collect the training history to compute the PV measurement. The PV measurement shows how the accuracy changes in response to the injected noise and indicates that overfitting is present if the accuracy does not decrease significantly on the noisy data. Werpachowski et al. (2019) generate adversarial examples to detect whether an image classification model is overfit to the test set. Chatterjee & Mishchenko (2020) describe a method that converts machine learning models to logic circuits and detects overfitting by inspecting rare patterns of handling training samples in the model. In this paper, we propose an overfitting detection method based on time series classifiers that only relies on the training history of a trained model and does not involve model conversion or retraining. We train the time series classifiers with training histories and labels indicating whether or not there is overfitting, hence, the classifiers can identify overfitting from the training history of a trained model. Since similar to our method, correlation-based methods detect overfitting based on the training history as well, we select them as the baseline to compare our method against and introduce it below.

**Correlation-based methods.** Inspired by the overfitting detection method of Kronberger et al. (2011), we compute correlation metrics between the training and validation loss to detect overfitting in DL models. The idea behind this method is intuitive: the training and validation loss (similar to the training and validation fitness in symbolic regression) are expected to be strongly correlated when there is no overfitting and the correlation should be weak when there is overfitting. The calculated correlation metrics are compared with a threshold (more details on how we select the threshold in Section 4) to determine if there is overfitting. We choose three correlation metrics: Spearman, Pearson (Hauke & Kossowski, 2011), and time-lagged Pearson correlation coefficients. We calculate both Spearman and Pearson correlation coefficients since we do not know whether the relationship between training and validation loss is linear. In addition, we compute the Pearson correlation coefficient between the time-lagged version (5-epoch lagged) of the training loss and the validation loss. This method is inspired by autocorrelation (Brockwell & Davis, 2002), which computes the correlation between a time series data and a time-lagged version of itself.

## 2.3 Overfitting prevention

Bejani & Ghatee (2021) identify three categories of overfitting prevention methods: *passive, active* and *semi-active methods*. *Passive methods* are employed before training a model, and include methods such as hyper-parameter optimization and model selection. For instance, Sun et al. (2017) improve the backpropagation algorithm to speed up the training process and avoid overfitting. Xu et al. (2021) introduce a learning algorithm based on a probabilistic model for avoiding overfitting to the noise in the training data. *Active methods* prevent overfitting by either imposing noise to the data or model through methods like adding dropout layers or other regularization schemes so that models cannot memorize the patterns in the data. For instance, Dropout (Srivastava et al., 2014), a simple and popular overfitting prevention method, randomly disables a part of the DL model during training to prevent overfitting. Finally, *semi-active methods* change the model architecture during the training process. They either work by adding hidden nodes or pruning existing nodes. All aforementioned overfitting prevention methods are typically intrusive (i.e., they require either modification of the model internals or data that is fed to the model) and require considerable expertise to accurately execute. For instance, using passive methods such as hyperparameter optimization to avoid overfitting requires expertise on the right optimization method to choose and the right parameters to tune, which is a vast area of research in itself (Bergstra & Bengio, 2012; Falkner et al., 2018; Bischl et al., 2021). Similarly, active and semi-active methods such as dropout or pruning require either adding layers or dynamically editing the model structure. In addition, even though these methods have been known to avoid overfitting they cannot guarantee that the model does not overfit and they typically employ methods like Early stopping (Morgan & Bourlard, 1989; Prechelt, 2012) to further predict if overfit might occur. Early stopping is a widely used overfitting prevention method (which we explain below) that is non-intrusive and does not require considerable expertise to execute.

**Early stopping.** Early stopping stops training when there is no improvement in a fixed number of epochs (indicated by the patience parameter) and returns the *best epoch* which has the lowest validation loss. The idea behind this method is that the training will converge or become overfit

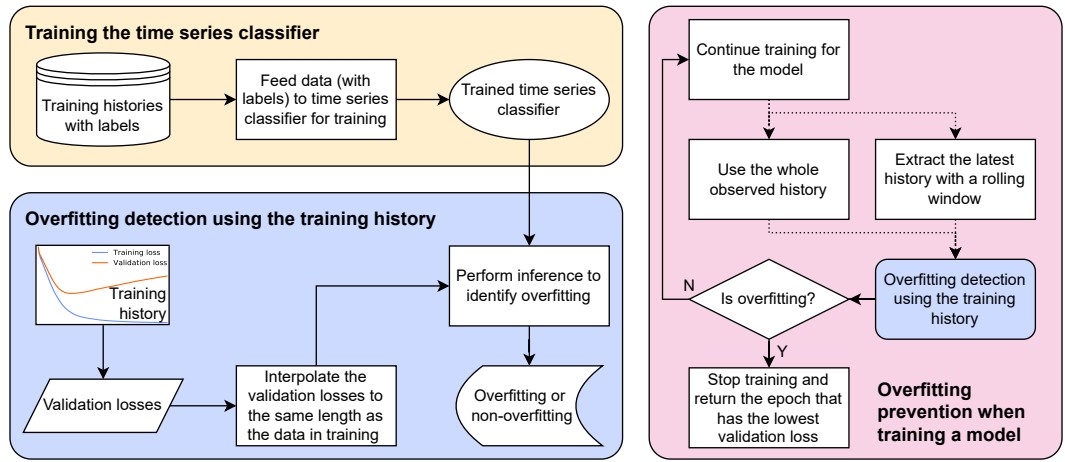

Figure 2: Our method for overfitting detection and prevention.

when the validation loss stops improving. However, Prechelt (2012) studied three stopping criteria of early stopping and found that using a slow stopping criterion will increase the training time while producing only a small improvement in generalization. In this paper, we propose using overfitting detection methods during the training process to prevent overfitting. Hence, our method prevents overfitting based on the byproduct (i.e., the training history) of the training process. We compare our method to the early stopping method, as both methods are non-intrusive and do not require significant expertise. We also include the results of an alternative version of early stopping based on smoothed validation loss in Appendix E.

## 3 OUR METHOD

Figure 2 shows an overview of our proposed method. Our method uses a time series classifier to detect and prevent overfitting. To the best of our knowledge, we are the first to use a time series classification-based method to detect and prevent overfitting. First, we collect a simulated dataset (more details on how we collect the data in Section 4) that contains training histories (i.e., training and validation loss curves, however we only use the validation loss curves in our method) with labels indicating whether overfitting occurs in order to train our time series classifier. Second, we train a time series classifier on all the training histories of the simulated dataset. We evaluate six state-of-the-art time series classifiers (see Appendix A) to identify the best-performing one. Finally, we use the trained time series classifier to perform both overfitting detection and prevention as follows.

**Overfitting detection**. To detect overfitting in a trained model, we first collect its validation losses over the training epochs. We feed this loss to our trained time series classifier to detect if there is overfitting. However, we cannot directly feed these validation losses to our classifier as the length of the validation losses might not be of the same length as that of the data used to train these time series classifiers. Except KNN-DTW all the studied time series classifiers expect the length of the inputs used for training for which the inference is made to be the same. Therefore, we first linearly interpolate the validation losses of the model for which we need to detect overfit to the same length as the training histories used to train the studied time series classifiers. We feed the interpolated validation losses to our trained time series classifier and perform inference to determine if the model is overfit. Figure 4 shows how the linear interpolation process works; if we only have validation losses over 8 epochs and our time series classifier was trained over 80 epoch validation loss values, we interpolate the 8 epoch losses to 80 so that we can feed it to the trained time series classifier.

**Overfitting prevention.** To prevent overfitting, we feed the training history (i.e., validation loss curve) of a DL model that is being trained to our trained time series classifier during the training process. The history is fed for inference in two different ways: (1) as a rolling window: we extract the latest history in a fixed window size (e.g., the latest 20 epochs), and (2) as the whole observed history (from the first to the latest epochs). Our time series classifier detects if in the fed history

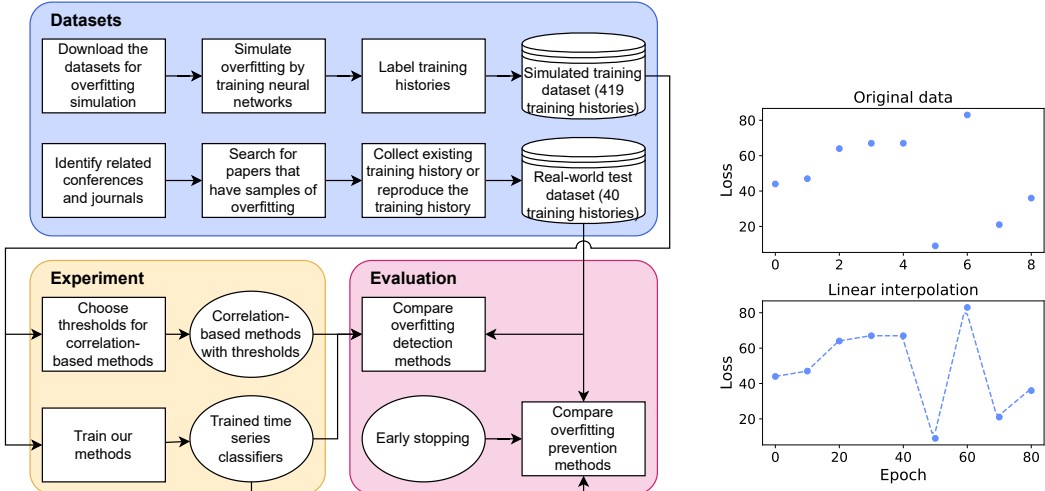

Figure 3: Overview of the experimental setup.

Figure 4: An example of linearly interpolation.

overfitting occurs. Similar to overfitting detection, we linearly interpolate the data before feeding it into our model. If there is no overfit occurring, we continue the training and repeat the above procedure until the model has finished training. For the rolling window, we move the window by a fixed step size and make another prediction. If our model detects the presence of overfit in the fed history, we return the lowest validation loss in the observed epochs as the best epoch.

## 4 EXPERIMENTAL SETUP

In this section, we introduce the datasets for training and evaluating the studied overfitting detection and prevention methods, the experiments of our study, and the evaluation metrics for the studied methods. Figure 3 shows an overview of the experimental setup.

### 4.1 DATASETS

**Simulated training dataset.** We use simulated training histories with labels to determine the threshold for the correlation-based methods (see Section 2.2) and to train our method as we explain in Section 3. We create our simulated dataset by training neural networks with different model complexities to generate the training history of overfitting and non-overfitting samples as follows:

*Step 1 – Download the datasets for overfitting simulation.* We download 12 datasets of real-world problems from the Proben1 (Prechelt et al., 1994) benchmark set for training neural networks (the information about the studied datasets can be found in Appendix A). These datasets were used by Prechelt (2012) to simulate training histories for studying early stopping. Furthermore, all of these datasets (except the "building" one) were originally collected from the UCI machine learning repository (Blake et al., 1998), which has been widely used in deep learning research (Kuleshov et al., 2018; Sajeev et al., 2019; Shi et al., 2021; Gadde et al., 2021). These datasets are pre-partitioned into training, validation, and test set (respectively 50%, 25%, and 25% of the data). In addition, Proben1 partitions each dataset three times in order to generate three distinct permutations. Hence, in total we collect 36 datasets from Proben1.

*Step 2 – Simulate overfitting by training neural networks.* We train Neural Networks (NNs) with various architectures on the collected 36 datasets. We do so to vary the model complexity which in turn increases the chance of producing an overfitted model, in the same way as Prechelt (2012) did in their study. The input/output layer contains the same number of nodes as the number of input/output coefficients of the datasets (see Appendix A) and rectified linear units (ReLUs) are used for all hidden layers. The structures of the NNs are as follows: (1) 6 one-hidden-layer NNs with hidden nodes of 2, 4, 8, 16, 24, 32, and (2) 6 two-hidden-layer NNs with hidden nodes (represented as first

layer hidden nodes + second layer hidden nodes) of 2+2, 4+2, 4+4, 8+4, 8+8, 16+8. We use the mean square error (MSE) as the loss function for regression problems, and cross entropy as the loss function for classification problems. Additionally, we used SGD as the optimizer for all of these problems. To increase the likelihood of overfitting, we train these 12 neural network architectures on each dataset (of the collected 36 datasets) for 1,000 epochs, producing 432 training histories.

*Step 3 – Label training histories.* To train our proposed method and the correlation-based methods, we need to manually label the training history as either "overfit", "non-overfit" or "uncertain". To ensure the robustness of our manual labelling process, we follow the approach outlined by Ding et al. (2020). The first and second authors of this paper independently labelled the 432 data points and discussed the results. In the first discussion round, the authors reached a 95% agreement (410 data points), and 10 data points that were labelled as "uncertain" by both authors were eliminated. In the second round, we discussed the 22 disagreements. Following the discussion, we eliminated 3 data points (labelled "uncertain" by both authors) and agreed on the labels for the remaining 19 data points. The final data set consists of 44 overfit and 375 non-overfit training histories. As an alternative (automated) approach for collecting the labels we experimented with a heuristic method, however, this method did not perform well (see Appendix H).

**Real-world test dataset.** To evaluate our method on real-world data, we surveyed papers from conferences and journals to collect samples of overfit and non-overfit models:

*Step 1 – Identify related conferences and journals.* We identify related conferences and journals based on the Computing Research and Education Association of Australasia (CORE[1]) and China Computer Federation (CCF[2]) ranking systems. Under the CCF A rank, we have 7 conferences and 4 journals in the "Artificial Intelligence" field. Under the CORE A* rank, we have 16 conferences in the "machine learning" and "artificial intelligence" fields and 12 journals in the "artificial intelligence and image processing" field. We get 17 conferences and 12 journals after merging the results because of the overlap between these two ranking systems.

*Step 2 – Search for papers that have samples of overfitting.* We found 33 full papers (see Appendix C) with the "overfit" keyword (which includes e.g., "overfitting") in the title that were published at the selected conferences and journals in the last 5 years. Five papers contain samples of overfitting: P2 - Chatterjee & Mishchenko (2020); P4 - Chen et al. (2021) P13 - Kim et al. (2021); P17 - Rice et al. (2020) and P23 - Singla et al. (2021). Appendix C lists the papers and the number of collected samples of overfitting (some of them also provide samples of non-overfitting).

*Step 3 – Collect existing training history or reproduce the training history.* Paper P17 shared the training history, making its replication straightforward. We replicated the other papers that provide overfitting samples to collect the training histories of these samples. We ran the code from the papers that provide replication packages (P4, P13, and P23) to generate the training history, but we were unable to replicate paper P13's results. We followed the methodology to replicate the results and training history for paper P2, which did not provide a replication package. In total, we collected 29 training histories of overfit models and 11 of non-overfit models (see Appendix C for details).

## 4.2 EXPERIMENTS

**Overfitting detection.** We train the time series classifiers based on the simulated dataset. We performed a grid search with 3-fold cross validation to tune the hyperparameters for each classifier based on the simulated dataset. After selecting the hyperparameters, we trained each time series classifier using the training histories and labels from the simulated dataset and saved the trained classifier. Furthermore, we search thresholds for the correlation-based methods on the simulated dataset. We perform a grid search for the thresholds between -1 and 1 based on the collected simulated dataset to select the threshold which has the best F-score.

**Overfitting prevention.** We reused the trained time series classifiers from the previous step to perform inference during the training process to prevent overfitting. We applied our method to the trained models in every 10 epochs (i.e., the step size) with 20, 40, 60, 80, and 100 epochs as different rolling window sizes. We set the patience values for early stopping from 5 to 115 epochs.

---

[1] https://www.core.edu.au/conference-portal
[2] https://ccf.atom.im/

Table 1: Results of the overfitting detection methods on the real-world dataset (Prec: precision; Rec: recall; F-s: F-score; Avg F-s: average F-score), and the time cost of training the studied methods on the simulated dataset and performing inference on the real-world dataset (per sample).

| Detection method | | Non-overfitting | | | Overfitting | | | Avg F-s | Training time (s) | Inference time (ms) |
|---|---|---|---|---|---|---|---|---|---|---|
| | | Prec | Rec | F-s | Prec | Rec | F-s | | | |
| Correlation based | Spearman | 0.71 | 0.91 | 0.80 | 0.96 | 0.86 | 0.91 | **0.86** | 2.461 | 0.908 |
| | Pearson | 0.78 | 0.64 | 0.70 | 0.87 | 0.93 | 0.90 | 0.80 | 0.222 | 0.025 |
| | Autocorr | 0.80 | 0.73 | 0.76 | 0.90 | 0.93 | 0.92 | 0.84 | 0.233 | 0.026 |
| **Time series classifier (ours)** | KNN-DTW | 0.79 | 1.00 | 0.88 | 1.00 | 0.90 | 0.95 | **0.91** | 0.001 | 180.512 |
| | HMM-GMM | 0.30 | 0.27 | 0.28 | 0.73 | 0.76 | 0.75 | 0.52 | 99.751 | 17.750 |
| | TSF | 0.77 | 0.91 | 0.83 | 0.96 | 0.90 | 0.93 | 0.88 | 0.311 | 17.209 |
| | TSBF | 0.79 | 1.00 | 0.88 | 1.00 | 0.90 | 0.95 | **0.91** | 0.301 | 31.683 |
| | BOSSVS | 0.46 | 0.91 | 0.61 | 0.94 | 0.59 | 0.72 | 0.67 | 1.877 | 19.342 |
| | SAX-VSM | 0.83 | 0.91 | 0.87 | 0.96 | 0.93 | 0.95 | **0.91** | 0.912 | 17.474 |

### 4.3 EVALUATION

**Evaluation metrics for overfitting detection.** To evaluate the classification performance of over-fitting detection methods, we calculated the **precision**, **recall**, and **F-score** for overfitting and non-overfitting samples in the real-world test dataset. In addition, we calculated the **average F-score** for directly comparing the classification performance of the studied methods. To evaluate the time cost of training and using the studied methods, we report the **training time** (in seconds) of each method on the simulated dataset and the **inference time** (in milliseconds) on the real-world dataset.

**Evaluation metrics for overfitting prevention.** Ideally, an overfitting prevention method returns the optimal epoch that has the optimal predictive performance for the model on the validation set and stops the training process as fast as possible. We define the **optimal rate** of an overfitting prevention method as the percentage of times the optimal epoch is identified. To evaluate the speed of the method, we define the **delay** as the epoch difference between the stopped epoch and the best epoch, e.g., the delay will be 10 epochs if a prevention method stops at the $123^{th}$ epoch while the $113^{th}$ epoch is the best one. For early stopping, the delay will be the same as the patience parameter.

## 5 RESULTS

**Overfitting detection.** *Our overfitting detection approach using time series classifiers (except HMM-GMM and BOSSVS) has a better classification performance than the correlation-based methods for overfitting detection.* Table 1 shows that our approach using KNN-DTW, TSBF, and SAX-VSM have the best F-score (0.91) on the real-world dataset, followed by TSF which outperforms the baseline methods as well. Even though our approach using BOSSVS achieves an F-score of 1 on the simulated dataset (see Appendix B), it performs poorly on the real-world dataset. From the results on both simulated and real-world datasets, the HMM-GMM time series classifier performs poorly for the problem of overfitting detection. We also note that the investigated correlation-based methods have a reasonably good performance. All of the studied correlation-based overfitting detection methods have F-scores above 0.8. However, our approach outperforms the best performing correlation-based overfitting detection approach by at least 5% on the studied real-world dataset. We also report the results of using perturbation validation for overfitting detection as an example of an intrusive overfitting detection method in Appendix D.

*The studied time series classifiers are more computationally intensive than correlation-based methods for inference, yet they are still useful in practice.* As shown in Table 1, our method requires more time for performing inference than the correlation-based methods. For instance, TSF has the fastest inference time among the classifiers but is around 20 times slower than the Spearman correlation-based method and around 700 times slower than the other two correlation-based methods. However, the speed of our method is not prohibitive in practice since overfitting detection is only executed once after the training is complete. It is also useful to note that the training times of the time series classifiers in our approach are not excessive. For instance, the training times of TSF and TSBF are around 300 milliseconds and KNN-DTW, our best performing time series classifier can finish train-

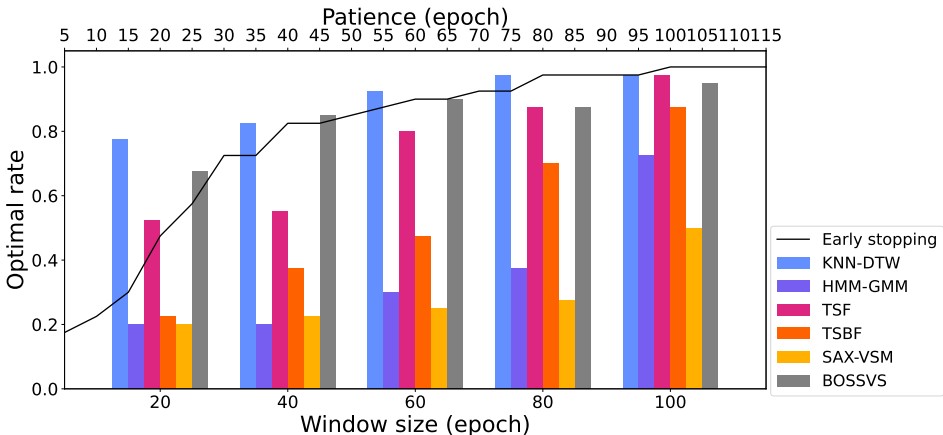

Figure 5: The optimal rate of our methods (using a rolling window) and early stopping with different patience values.

ing in 1 millisecond. However, KNN-DTW requires the longest time for inference which is around 180 milliseconds for a training history.

**Overfitting prevention.** *Our method with KNN-DTW (both using rolling window and whole observed history) is often more accurate than early stopping for overfitting prevention.* Other studied classifiers do not perform as well as KNN-DTW for overfitting prevention. As Figure 5 and Table 2 show, both our approach based on rolling window and whole observed history has higher accuracy than early stopping at identifying the optimal epoch. In particular, our approach with KNN-DTW based on rolling window is more accurate (see Figure 5) than early stopping when using up to 80 epochs as the patience parameter and window size. For example, our method with KNN-DTW obtains 78% accuracy when setting the window size to 20 epochs, while early stopping has 48% accuracy when setting the patience parameter to that number of epochs. However, we find that early stopping achieves nearly perfect accuracy when the patience is larger than 80 epochs. The reason is that 90% of the training histories in the real-world dataset have around 200 epochs, hence, a large patience value makes it easy for early stopping to choose the optimal epoch. In addition, Table 2 shows that our method with KNN-DTW based on the whole observed history also obtains higher accuracy than early stopping. For instance, the KNN-DTW classifier has 95% accuracy with a median delay of 43.5 epochs while early stopping has only 83% using the same number (i.e., between 40 to 45 epochs patience in Figure 5).

Table 2: The optimal rate and median delay of our overfitting prevention methods that are based on the whole observed history.

| Classifier | Optimum rate | Median delay |
|---|---|---|
| KNN-DTW | 0.95 | 43.5 |
| HMM-GMM | 0.18 | 0.0 |
| TSF | 0.90 | 35.0 |
| TSBF | 0.83 | 31.0 |
| BOSSVS | 0.65 | 21.0 |
| SAX-VSM | 0.33 | 10.0 |

Table 3: The median delay of our overfitting prevention methods that are based on the rolling window with different window sizes.

| Classifier | Window size (epoch) | | | | |
|---|---|---|---|---|---|
| | 20 | 40 | 60 | 80 | 100 |
| KNN-DTW | 31.0 | 27.0 | 37.5 | 42.5 | 45.5 |
| HMM-GMM | 5.0 | 6.5 | 16.5 | 28.0 | 41.5 |
| TSF | 12.5 | 22.0 | 31.0 | 39.5 | 44.0 |
| TSBF | 8.5 | 15.0 | 25.0 | 37.0 | 47.0 |
| BOSSVS | 7.0 | 29.0 | 34.5 | 48.5 | 56.5 |
| SAX-VSM | 4.0 | 11.0 | 9.5 | 16.0 | 24.0 |

*Our method using KNN-DTW and a rolling window can stop training a DL model earlier than early stopping.* As shown in Table 3, with the same number of epochs for the patience parameter and window size, our method with KNN-DTW based on the rolling window (except window size 20) can save training time (i.e., there is a smaller delay between the stopped epoch and the best epoch) over early stopping. For instance, when setting both the patience parameter and window size to 40 epochs, KNN-DTW and early stopping have the same accuracy and KNN-DTW has a

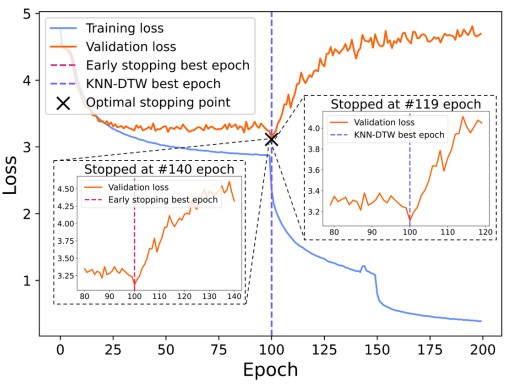 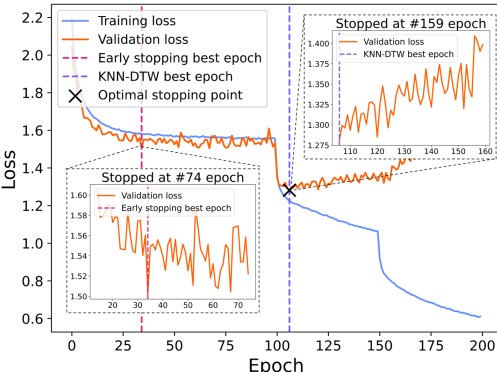

(a) Our method stops earlier than the early stopping but both achieve the same optimal epoch.

(b) Our method stops later than the early stopping but achieves the optimal epoch.

Figure 6: Examples of overfitting prevention based on KNN-DTW (set the window size as 40 epochs) and early stopping (set the patience parameter as 40 epochs).

median delay of 27 epochs while early stopping has a fixed delay of 40 epochs (which is the same as the patience parameter). In comparison to the delay in early stopping, the delay between the stopped epoch and the best epoch is at least 32% shorter with our method using KNN-DTW (see Appendix F for the significance testing results). Figure 6a shows an example in which early stopping and our method both identify the optimal epoch, but our method stops 21 epochs earlier than early stopping (which stops with a 40 epochs delay). In addition, our method does not sacrifice accuracy for a shorter delay (as shown in Figure 5). Figure 6b shows an example in which our method stops later than early stopping but identifies the optimal epoch when using the same number of epochs for the patience parameter and window size. Furthermore, our overfitting detection approach can be used with zero-one validation loss; the results of our approach and early stopping with zero-one validation loss can be found in Appendix G.

*Among our two approaches for overfitting prevention we recommend the usage of KNN-DTW with rolling window.* Though using the whole observed history may be more accurate at predicting the optimal epoch than using a rolling window for our approach, we note that we can predict the optimal epoch much earlier with the rolling window approach for a very small trade-off in accuracy. As shown in the Table 3 and Figure 5, our method with KNN-DTW achieves 83% accuracy with a median delay of 27 epochs and 90% accuracy with a median delay of 37.5 epochs using the window size as 40 and 60 epochs respectively. However, the median delay of KNN-DTW when using whole observed history is 43.5 while using the rolling window with a window size of 80 or more epochs can achieve a higher accuracy (98% vs. 95% accuracy) with a shorter delay (42.5 vs. 43.5 epochs). In summary, we suggest using the rolling window approach since it is stops earlier with a relatively small accuracy drop using a small window (e.g., 40 epochs) and performs better than the observed whole history approach when using a large window size (e.g., 80 epochs).

## 6 CONCLUSION AND FUTURE WORK

We propose a non-intrusive overfitting detection and prevention method that is based on time series classifiers trained on the training history of DL models. Our method (when using the KNN-DTW time series classifier) has (1) better classification performance than correlation-based methods for overfitting detection, and (2) greater accuracy than early stopping for overfitting prevention with a shorter delay. We do so using a real-world dataset of labelled training histories collected from the papers published at top AI venues in the last 5 years. Furthermore, the trained time series classifiers are included in our replication package for use by other researchers. One of the downsides of our approach is that our best performing time series classifier takes longer to perform the inference required to detect and prevent overfit than the studied baselines. We encourage future work to optimize time series classifiers to enable overfitting detection and prevention in real-time with smaller delays.

## Reproducibility Statement

The replication package can be found at: `https://github.com/anonymous-p/overfit_detect`. This package includes the code for training and using the studied methods, notebooks for analysing the results, the simulated and real-world datasets, and the trained models. Furthermore, we are currently developing a tool to use the studied methods and will make it public once completed.

**Experimental environment.** We use Python 3.8 with TensorFlow 2.9.0 and run experiments on Ubuntu 20.04 with Linux kernel 5.15.0. The hardware specifications are as follows: (1) NVIDIA RTX 3090 GPU with 24 GB memory (the versions of CUDA and cuDNN are 10.1.243 and 7.6.5), (2) 3.50 GHz Intel(R) Core(TM) i9-11900K CPU, and (3) 64 GB RAM.

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

# A  STUDIED DATASETS AND TIME SERIES CLASSIFIERS

Table 4 shows the information about the studied datasets. There are 3 datasets for regression tasks and 9 datasets for classification tasks.

Table 4: Information about studied datasets.

| Dataset | Type | #In | #Out | #Examples |
|---|---|---|---|---|
| building | regression | 14 | 3 | 4,208 |
| cancer | classification | 9 | 2 | 699 |
| card | classification | 51 | 2 | 690 |
| diabetes | classification | 8 | 2 | 768 |
| flare | regression | 24 | 3 | 1,066 |
| gene | classification | 120 | 3 | 3,175 |
| glass | classification | 9 | 6 | 214 |
| heart | classification | 35 | 2 | 920 |
| hearta | regression | 35 | 1 | 920 |
| horse | classification | 58 | 3 | 364 |
| soybean | classification | 82 | 19 | 683 |
| thyroid | classification | 21 | 3 | 7,200 |

Table 5 introduces the studied time series classifiers. Since there has been no prior systematic research on time series classifiers for training histories as a reference, we choose the classifiers that have been reported as baselines or state-of-the-art in studies (Xi et al., 2006; Varol et al., 2017; Anami & Bhandage, 2019; Wang et al., 2021)

Table 5: Studied time series classifiers.

| Classifier | Description |
|---|---|
| KNN-DTW[*] | Uses K-Nearest Neighbors (Hand, 2007) and Dynamic Time Warping (Ding et al., 2008) as the distance metric |
| HMM-GMM | Uses Hidden Markov Model for modeling time series data and Gaussian Mixture Model as the emissions probability density (Gauvain & Lee, 1994; Ji et al., 2006) |
| TSF | Uses a random forest (Biau & Scornet, 2016) for time series data using an ensemble of time series trees (Deng et al., 2013) |
| TSBF | Time Series Bag-of-Features (Baydogan et al., 2013) extracts features based on the bag-of-features approach (Fu et al., 2011) to create a random forest |
| SAX-VSM | Symbolic Aggregate approXimation transforms the data into symbolic representations (Lin et al., 2007) and Vector Space Model (Salton et al., 1975; Peng et al., 2014) transforms them into vectors to calculate similarity for classification |
| BOSSVS | Bag-of-SFA Symbols in Vector Space (Schäfer, 2016) is similar to SAX-VSM but use SFA (Schäfer & Högqvist, 2012) to transform the data instead of SAX |

[*] We do not interpolate the validation loss for KNN-DTW since it does not require a constant length input signal.

# B  RESULTS OF TRAINING OVERFITTING DETECTION METHODS

As mentioned in Section 4.2, we tune the hyperparameters of the time series classifiers by performing a grid search with 3-fold cross validation. Table 6 shows that all the classifiers achieve F-scores of more than 0.95 in the 3-fold cross validation except HMM-GMM which obtains only an average F-score of only 0.6. Table 7 shows the results of training correlation-based methods and our method on the simulated dataset. Generally, the time series classifiers achieve better performance on the simulated dataset than the other methods (except the HMM-GMM classifier) and three of the classifiers can even correctly identify all the data in the simulated dataset.

We notice that KNN-DTW performs well on both the simulated and real-world datasets. One possible reason for the performance of KNN-DTW might be that DTW is good at measuring similarity

Table 6: The F-scores of the time series classifiers in 3-fold cross validation (CV) on the simulated dataset.

| Classifier | CV1 | CV2 | CV3 | Avg | Variance |
|---|---|---|---|---|---|
| KNN-DTW | 0.98 | 0.98 | 0.96 | 0.97 | 0.00 |
| HMM-GMM | 0.41 | 0.38 | 1.00 | 0.59 | 0.08 |
| TSF | 0.98 | 0.98 | 1.00 | 0.99 | 0.00 |
| TSBF | 0.98 | 0.98 | 1.00 | 0.99 | 0.00 |
| BOSSVS | 1.00 | 1.00 | 1.00 | 1.00 | 0.00 |
| SAX-VSM | 0.96 | 0.96 | 0.96 | 0.96 | 0.00 |

Table 7: Results of overfitting detection methods on the simulated dataset. (Prec: precision; Rec: recall; F-s: F-score; Avg F-s: macro average F-score)

| Detection method | | Non-overfitting | | | Overfitting | | | Avg F-s |
|---|---|---|---|---|---|---|---|---|
| | | Prec | Rec | F-s | Prec | Rec | F-s | |
| Correlation based | Spearman | 0.99 | 1.00 | 0.99 | 0.95 | 0.89 | 0.92 | **0.95** |
| | Pearson | 0.99 | 0.97 | 0.98 | 0.79 | 0.93 | 0.85 | 0.92 |
| | Autocorr | 0.97 | 0.94 | 0.95 | 0.59 | 0.73 | 0.65 | 0.80 |
| **Time series classifier (ours)** | KNN-DTW | 0.99 | 1.00 | 1.00 | 0.98 | 0.93 | 0.95 | 0.97 |
| | HMM-GMM | 0.95 | 0.55 | 0.70 | 0.16 | 0.75 | 0.27 | 0.48 |
| | TSF | 1.00 | 1.00 | 1.00 | 1.00 | 1.00 | 1.00 | **1.00** |
| | TSBF | 1.00 | 1.00 | 1.00 | 1.00 | 1.00 | 1.00 | **1.00** |
| | BOSSVS | 1.00 | 1.00 | 1.00 | 1.00 | 1.00 | 1.00 | **1.00** |
| | SAX-VSM | 0.99 | 1.00 | 0.99 | 0.98 | 0.91 | 0.94 | 0.97 |

across curves, which aids KNN in distinguishing between overfit and non-overfit samples. In contrast, HMM-GMM performs poorly on both the simulated training and real-world test datasets. One possible explanation is that the extracted state models (via HMM) of the curves do not follow a Gaussian probability distribution. BOSSVS correctly identifies all the data in the simulated dataset but performs poorly on the real-world dataset (as shown in Table 1), which may be due to overfitting the simulated dataset.

## C    SURVEYED PAPERS

Table 8 lists the 33 papers we surveyed from top AI conferences and journals. We found 5 papers out of the 33 papers provide samples of overfit and fit models and collected 40 samples from these papers (as shown in Table 9).

Table 8: Information about the surveyed papers.

| #P | Authors | Title | Venue | Year |
|---|---|---|---|---|
| P1 | Belkin et al. | Overfitting or perfect fitting? risk bounds for classification and regression rules that interpolate. | NeurIPS | 2018 |
| P2 | Chatterjee & Mishchenko | Circuit-based intrinsic methods to detect overfitting. | ICML | 2020 |
| P3 | Chatterji & Long | Foolish crowds support benign overfitting. | JMLR | 2022 |
| P4 | Chen et al. | Robust overfitting may be mitigated by properly learned smoothening. | ICLR | 2021 |
| P5 | d'Ascoli et al. | Triple descent and the two kinds of overfitting: where & why do they appear? | NeurIPS | 2020 |
| | | | Continued on next page | |

Table 8 – continued from previous page

| #P | Authors | Title | Venue | Year |
|---|---|---|---|---|
| P6 | Feldman et al. | The advantages of multiple classes for reducing overfitting from test set reuse. | ICML | 2019 |
| P7 | Feldman et al. | Open problem: how fast can a multiclass test set be overfit? | COLT | 2019 |
| P8 | Frei et al. | Benign overfitting without linearity: neural network classifiers trained by gradient descent for noisy linear data. | COLT | 2022 |
| P9 | He et al. | Sparse double descent: where network pruning aggravates overfitting. | ICML | 2022 |
| P10 | Huang et al. | Sparse progressive distillation: resolving overfitting under pretrain-and-finetune paradigm. | ACL | 2022 |
| P11 | Ju et al. | Overfitting can be harmless for basis pursuit, but only to a degree. | NeurIPS | 2020 |
| P12 | Ju et al. | On the generalization power of overfitted two-layer neural tangent kernel models. | ICML | 2021 |
| P13 | Kim et al. | Understanding catastrophic overfitting in single-step adversarial training. | AAAI | 2021 |
| P14 | Koehler et al. | Uniform convergence of interpolators: Gaussian width, norm bounds and benign overfitting. | NeurIPS | 2021 |
| P15 | Liu et al. | Overfitting the data: compact neural video delivery via content-aware feature modulation. | ICCV | 2021 |
| P16 | Mohammed & Cawley | Over-fitting in model selection with Gaussian process regression. | ICML | 2017 |
| P17 | Rice et al. | Overfitting in adversarially robust deep learning. | ICML | 2020 |
| P18 | Roelofs et al. | A meta-analysis of overfitting in machine learning. | NeurIPS | 2019 |
| P19 | Rozendaal et al. | Overfitting for fun and profit: instance-adaptive data compression. | ICLR | 2021 |
| P20 | Russo & Zou | How much does your data exploration overfit? controlling bias via information usage. | IEEE Trans. Inf. Theory | 2020 |
| P21 | Sanyal et al. | How benign is benign overfitting? | ICLR | 2021 |
| P22 | Shamir | The implicit bias of benign overfitting. | COLT | 2022 |
| P23 | Singla et al. | Low curvature activations reduce overfitting in adversarial training. | ICCV | 2021 |
| P24 | Song et al. | Observational overfitting in reinforcement learning. | ICLR | 2020 |
| P25 | Steck | Autoencoders that don't overfit towards the identity. | NeurIPS | 2020 |
| P26 | Sun et al. | meProp: sparsified back propagation for accelerated deep learning with reduced overfitting. | ICML | 2017 |
| P27 | Telgarsky | Stochastic linear optimization never overfits with quadratically-bounded losses on general data. | COLT | 2022 |
| P28 | Wang et al. | Benign overfitting in multiclass classification: all roads lead to interpolation. | NeurIPS | 2021 |
| P29 | Webster et al. | Detecting overfitting of deep generative networks via latent recovery. | CVPR | 2019 |
| P30 | Werpachowski et al. | Detecting overfitting via adversarial examples. | NeurIPS | 2019 |
| P31 | Xu et al. | Overfitting avoidance in tensor train factorization and completion: prior analysis and inference. | ICDM | 2021 |
| P32 | Zhang & Amini | Label consistency in overfitted generalized k-means. | NeurIPS | 2021 |
| P33 | Zhang et al. | Why overfitting isn't always bad: retrofitting cross-lingual word embeddings to dictionaries. | ACL | 2020 |

Table 9: Information about collected samples from surveyed papers.

| Paper | Labels for the training history in the manuscript | #Overfit | #Non-overfit |
|-------|---------------------------------------------------|----------|--------------|
| P2 | "[...] the validation accuracy of nn-random is 9.73% (i.e., close to chance) confirming that it is horribly overfit" | 2 | 0 |
| P4 | "We first observe that the robust overfitting prevails in all Baseline cases" "Our methods effectively mitigates the robust overfitting" | 3 | 3 |
| P13 | "Figure 4 [...] that is, catastrophic overfitting occurs." "Figure 6 shows that the proposed method also successfully prevents catastrophic overfitting [...]" | N/A* | N/A* |
| P17 | "Figure 24 [...] We see clear robust overfitting for the smaller two options in $\lambda$, and find no overfitting but highly regularized models for the larger two options [...]" | 20 | 4 |
| P23 | "These results therefore validate our claim that low curvature activations reduce robust overfitting" | 4 | 4 |

*: Cannot reproduce the same results as the paper.

Table 10: Results of the Perturbation Validation method for overfitting detection methods on the simulated dataset.

| Non-overfitting | | | Overfitting | | | Average |
|-----------|--------|---------|-----------|--------|---------|---------|
| Precision | Recall | F-score | Precision | Recall | F-score | F-score |
| 0.89 | 0.85 | 0.87 | 0.07 | 0.09 | 0.08 | 0.47 |

## D  PERTURBATION VALIDATION FOR OVERFITTING DETECTION

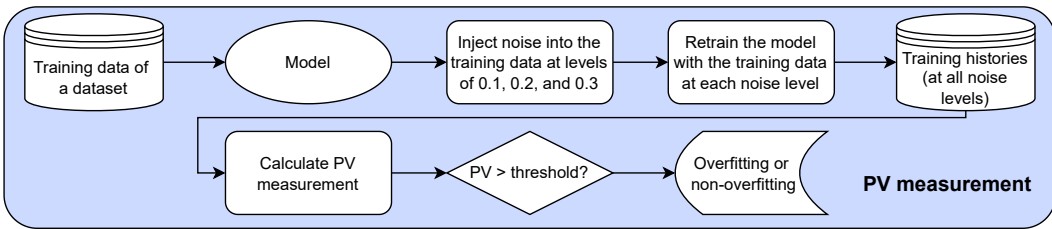

Figure 7: Our method for overfitting detection and prevention.

Zhang et al. (2019) suggest the perturbation validation (PV) assessment to determine whether a model fits the training data properly (i.e., ensure that it is neither overfitting nor underfit). As shown in Figure 7, We inject three levels of noise (0.1, 0.2, and 0.3) as the original paper (Zhang et al., 2019) into the labels in the training set and retrain the model. We repeat the training for each noise level and collect the training history to compute the PV measurement. The proposed PV measurement will show how much the accuracy decreases in response to the injected noise and indicate if overfitting is present. The idea behind this method is that overfit or underfit models would lose accuracy more slowly when trained using the noise-injected training set than optimally-fitted models. Since the calculated PV measurement is only one value and we compare it with a threshold to determine if there is overfitting. To select the threshold, we performed a grid search for the thresholds between -1 and 1 based on the F-score on the simulated dataset.

The PV measurement requires retraining the models, which takes a significant amount of computational time but results in poor performance in the simulated dataset (see Table 10). We do not calculate the PV measurement on the real-world dataset since retraining the models takes too long, and the performance on the simulated dataset does not justify this extra effort.

Table 11: Results of early stopping based on the smoothed validation loss curves.

| Patience | Smooth epochs | Optimum rate | Median delay | Patience | Smooth epochs | Optimum rate | Median delay |
|---|---|---|---|---|---|---|---|
| 20 | 0 | 0.48 | 20.0 | 60 | 0 | 0.90 | 60.0 |
| | 5 | 0.38 | 20.0 | | 5 | 0.93 | 61.0 |
| | 10 | 0.40 | 20.0 | | 10 | 0.93 | 62.0 |
| | 15 | 0.55 | 22.0 | | 15 | 0.90 | 62.0 |
| 40 | 0 | 0.83 | 40.0 | 80 | 0 | 0.98 | 80.0 |
| | 5 | 0.85 | 42.0 | | 5 | 0.98 | 81.0 |
| | 10 | 0.83 | 43.5 | | 10 | 0.98 | 82.0 |
| | 15 | 0.88 | 44.5 | | 15 | 0.90 | 81.5 |

## E  EARLY STOPPING BASED ON SMOOTHED VALIDATION LOSS CURVES

An alternate version of early stopping inspects the moving average of the smoothed validation loss curves (Shumway & Stoffer, 2017; Molugaram & Rao, 2017) to determine whether to stop the training process. After terminating the training process, early stopping returns the best epoch which has the lowest validation loss (not the smoothed value). Table 11 shows that using a smoothed validation loss curve may increase the optimal rate and slightly increases the delay. However, it does not have better performance than the basic form of early stopping and cannot compete with our approach. Figure 8 shows an example in which early stopping achieves the optimal epoch after using the smoothed validation loss. However, the smoothed curves can also hurt the performance, such as dropping 10% of the optimal rate when early stopping using a smoothing window of 5 epochs and a patience parameter of 20 epochs. Figure 9 illustrates an example of missing the optimal epoch after smoothing the validation loss curve.

## F  SIGNIFICANCE TEST FOR THE RESULTS OF OVERFITTING PREVENTION

To study the difference in delay across overfitting prevention approaches, we performed the Mann-Whitney U test (Mann & Whitney, 1947) at a significance level of $\alpha = 0.05$ to determine whether the distributions of the delay epochs of early stopping and our approach are significantly different. We also computed Cliff's delta $d$ (Long et al., 2003) effect size to quantify the difference based on the provided thresholds (Romano et al., 2006):

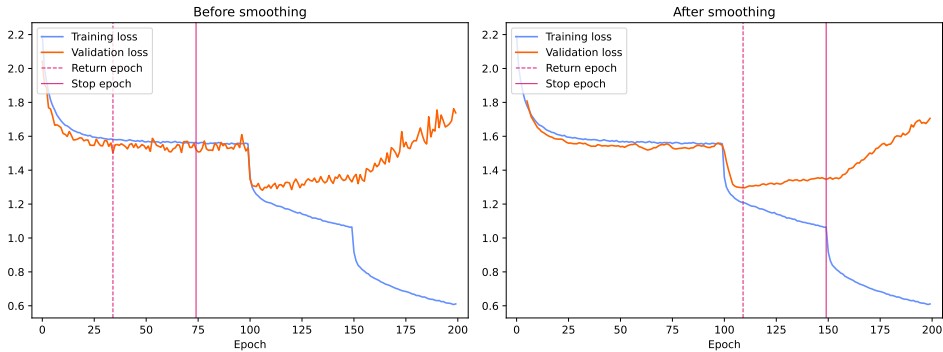

Figure 8: An example in which early stopping achieves the optimal epoch based on smoothed validation loss (the patience parameter is set to 40 epochs and the smoothing window size is 5).

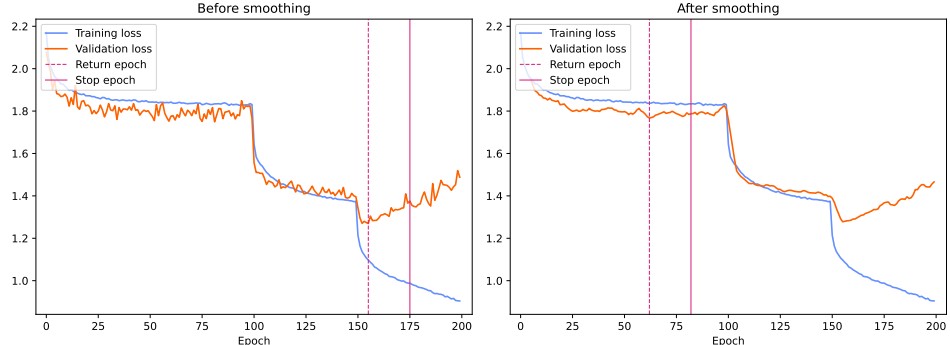

Figure 9: An example in which early stopping cannot achieve the optimal epoch based on smoothed validation loss (the patience parameter is set to 20 epochs and the smoothing window size is 5).

Table 12: Significance testing results of the delays for overfitting prevention. (Ws.: Window size)

| Classifier | Ws. | P | Effect Size | Cliff's d value | Classifier | Ws. | P | Effect Size | Cliff's d value |
|---|---|---|---|---|---|---|---|---|---|
| BOSSVS | 20 | 0.000 | large | -0.600 | SAX-VSM | 20 | 0.000 | large | -1.000 |
| | 40 | 0.001 | medium | -0.405 | | 40 | 0.000 | large | -0.956 |
| | 60 | 0.000 | large | -0.476 | | 60 | 0.000 | large | -0.868 |
| | 80 | 0.000 | large | -0.522 | | 80 | 0.000 | large | -0.817 |
| | 100 | 0.001 | medium | -0.449 | | 100 | 0.000 | large | -0.681 |
| HMM-GMM | 20 | 0.000 | large | -1.000 | TSBF | 20 | 0.000 | large | -0.875 |
| | 40 | 0.000 | large | -0.966 | | 40 | 0.000 | large | -0.893 |
| | 60 | 0.000 | large | -0.859 | | 60 | 0.000 | large | -0.742 |
| | 80 | 0.000 | large | -0.793 | | 80 | 0.000 | large | -0.748 |
| | 100 | 0.000 | large | -0.633 | | 100 | 0.000 | large | -0.581 |
| KNN-DTW | 20 | 0.101 | small | 0.200 | TSF | 20 | 0.000 | large | -0.525 |
| | 40 | 0.001 | medium | -0.392 | | 40 | 0.000 | large | -0.804 |
| | 60 | 0.000 | large | -0.485 | | 60 | 0.000 | large | -0.700 |
| | 80 | 0.000 | large | -0.541 | | 80 | 0.000 | large | -0.709 |
| | 100 | 0.000 | large | -0.517 | | 100 | 0.000 | large | -0.619 |

$$\text{Effect size} = \begin{cases} negligible, & \text{if } |d| \leq 0.147 \\ small, & \text{if } 0.147 < |d| \leq 0.33 \\ medium, & \text{if } 0.33 < |d| \leq 0.474 \\ large, & \text{if } 0.474 < |d| \leq 1 \end{cases} \tag{1}$$

Table 12 shows that the distributions of delay of early stopping and our methods are significantly different except for KNN-DTW with a window size of 20 epochs. In addition, the delays of our methods are shorter than early stopping (except for KNN-DTW with a window size of 20 epochs) with at least a medium effect size.

## G  USING ZERO-ONE LOSS FOR OVERFITTING PREVENTION

Overfitting prevention methods may stop the training process by inspecting the validation accuracy error (i.e., zero-one loss) rather than the loss utilized for optimizing the model. Figure 10 shows the optimal rate of our approaches as well as early stopping based on zero-one validation loss. When compared to the results in Figure 5 (which uses validation loss), the optimal rate of early stopping based on zero-one loss is quite similar. The results show that the optimal rate improves (with an average of 2.5%) when the patience value is between 35 and 90 epochs but declines (with an

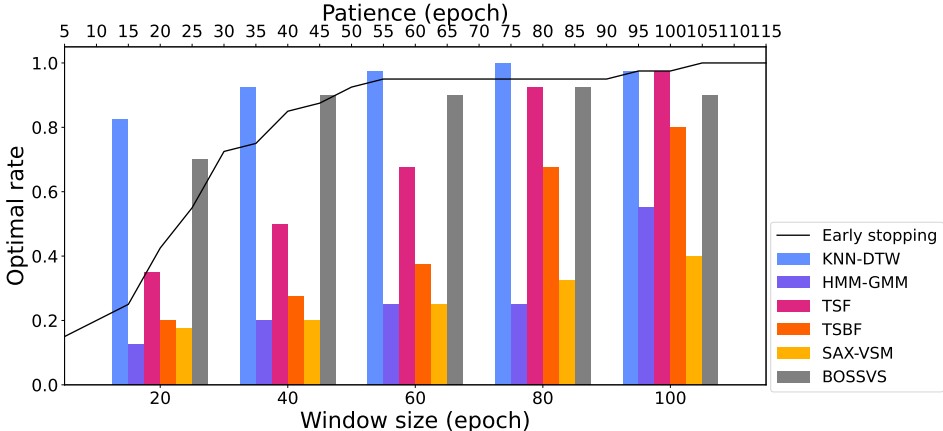

Figure 10: The optimal rate of our methods (using a rolling window) and early stopping with different patience values based on zero-one validation loss curves.

average of 2.9%) when the patience value is less than 35 epochs. For our method with KNN-DTW, the results show that using zero-one loss curves increases the optimal rate, which still outperforms early stopping and other time series classifiers. However, the delay of the KNN-DTW also increases and stops later than early stopping when the window size is less than or equal to 40 epochs (please see Table 13 for details). Furthermore, we found that our approach with KNN-DTW has a higher optimal rate but longer delay while using the zero-one loss curves (compared to using the original loss curves).

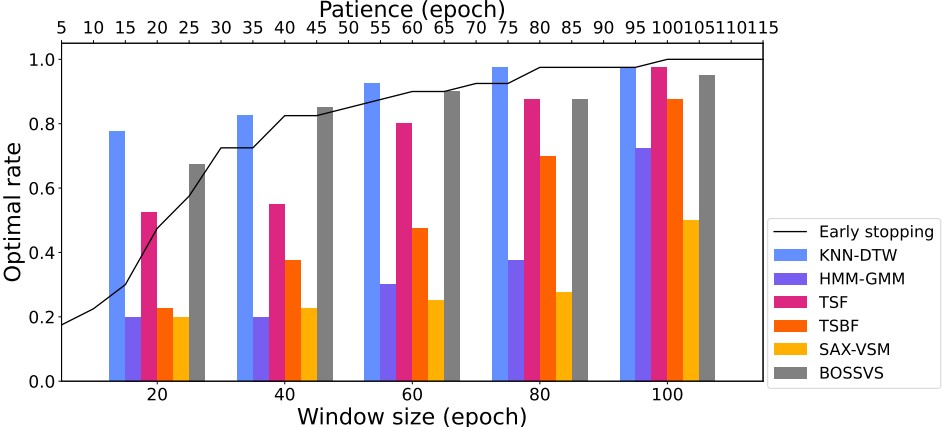

Figure 11: The optimal rate of our methods (using a rolling window) and early stopping with different patience values. (This figure is the same as Figure 5, we added it here for easy comparison to Figure 10.)

## H  A HEURISTIC METHOD FOR AUTOMATIC LABELLING

We developed a heuristic method for labelling the training history in the simulated dataset as over-fitting based on the following conditions:

- The training loss and validation loss both decrease in the first $inc_p$ percentage of the training history.
- The training loss and validation loss both decrease in the last $dec_p$ percentage of the training history.

Table 13: The median delay and significant testing of our overfitting prevention methods based on zero-one validation loss curves with different window sizes. (Ws.: Window size; Md.: Median delay)

| Ws. | Classifier | Md. | P | Effect size | Cliff's d value | Classifier | Md. | P | Effect size | Cliff's d value |
|---|---|---|---|---|---|---|---|---|---|---|
| 20 | | 13.5 | 0.684 | neg | -0.050 | | 5.0 | 0.000 | large | -0.950 |
| 40 | | 46.0 | 0.109 | small | 0.198 | | 5.5 | 0.000 | large | -0.976 |
| 60 | BOSSVS | 44.5 | 0.494 | neg | -0.087 | SAX-VSM | 11.5 | 0.000 | large | -0.884 |
| 80 | | 46.5 | 0.115 | small | -0.202 | | 14.5 | 0.000 | large | -0.806 |
| 100 | | 49.5 | 0.029 | small | -0.284 | | 26.5 | 0.000 | large | -0.716 |
| 20 | | 2.0 | 0.000 | large | -0.850 | | 7.5 | 0.000 | large | -0.750 |
| 40 | | 5.0 | 0.000 | large | -0.739 | | 14.5 | 0.000 | large | -0.867 |
| 60 | HMM-GMM | 8.0 | 0.000 | large | -0.656 | TSBF | 27.0 | 0.000 | large | -0.809 |
| 80 | | 18.0 | 0.000 | large | -0.719 | | 37.0 | 0.000 | large | -0.708 |
| 100 | | 46.5 | 0.001 | medium | -0.431 | | 45.0 | 0.000 | large | -0.561 |
| 20 | | 48.5 | 0.000 | large | 0.650 | | 12.5 | 0.001 | medium | -0.400 |
| 40 | | 44.0 | 0.027 | small | 0.273 | | 24.0 | 0.000 | large | -0.726 |
| 60 | KNN-DTW | 47.0 | 0.392 | neg | -0.109 | TSF | 31.0 | 0.000 | large | -0.630 |
| 80 | | 48.5 | 0.194 | small | -0.166 | | 42.5 | 0.000 | large | -0.596 |
| 100 | | 56.5 | 0.024 | small | -0.292 | | 47.0 | 0.000 | large | -0.479 |

- The gap between the training loss and validation loss exceeds $gap_p$ percentage of the sum of the training and validation loss.

To select the thresholds, we performed a grid search between 10% to 50% for $inc_p$ and $dec_p$, and a grid search between 1% to 50% for $gap_p$. The best performance of the heuristic method can achieve is a 0.75 F-score for the overfitting samples with 0.96 precision and 0.61 recall. The result shows that the heuristic method does not work well on the simulated dataset in comparison to other methods (see Table 7). Furthermore, the heuristic method performs poorly on the real-world dataset, with a 0.22 average F-score. Hence, human labels are still required for training the time series classifiers.

