# OpenReview forum: "Using the Training History to Detect and Prevent Overfitting in Deep Learning Models"
_ICLR.cc/2023/Conference — Submitted to ICLR 2023_

### Official Review · Reviewer_xGGx · 2022-10-16

**Confidence:** 3
**Correctness:** 2
**Technical Novelty And Significance:** 3
**Empirical Novelty And Significance:** 3
**Recommendation:** 5

**Clarity, Quality, Novelty And Reproducibility:**

Clarity: The paper is clear in most parts, but I had a hard time understanding how the labels are collected, especially on how they define the overfit/non-overfit labels (I wrote the details in the previous section).

Quality: I feel having more discussions on the type of overfitting the paper wants to address would raise the quality.

Novelty: The paper uses the training history to detect and prevent overfitting, which seems novel. However, the paper does not discuss other papers that utilize the training history, e.g., see the survey paper  "Learning Curves for Decision Making in Supervised Machine Learning - A Survey".

Reproducibility: A link to an anonymous github repo was provided in the paper. However, I did not check the code.

**Strength And Weaknesses:**

Strengths:
- Using the training history to detect and prevent overfitting is interesting.
- The paper constructs a new dataset of overfitting/non-overfitting samples by going through many recent ML papers on overfitting.
- The experiments show that detecting overfitting is possible, and that the time-series classifiers tends to perform better than the correlation based ones.
- Practical value with experiments showing that the method can find the optimal stopping point and avoid overfitting at least 32% earlier than early stopping and achieve at least the same accuracy as early stopping.

Weaknesses:
- If I understood the paper correctly, I feel the main issue is that the paper does not directly discuss what "overfitting" is. There are two sentences that try to address what overfitting is, in the abstract and introduction: 1) the abstract mentions that "overfitting of deep learning models on training data leads to poor generalizability on unseen data", and 2) 1st paragraph of Section 1 (Introduction) mentions that "for the overfit model, after a certain amount of training, the validation loss begins to increase while the training loss continues to decrease". The 1st one (in the abstract) can be regarded as a consequence for overfitting rather than an explanation of overfitting, so for example, the same sentence can be used for underfitting. The 2nd one (in the 1st paragraph of Section 1) is more specific, and if we use this as the definition of overfitting, it may exclude some learning curves which we may sometimes regard as overfitting or include ones that may not be overfitting. For example, if the validation loss stops decreasing after a certain number of epochs (but does not increase, just becomes flat) while the training loss keeps decreasing, the gap between training/validation loss will widen, and some people will regard this is a case of overfitting. As an extreme example, Fig 1 (b) can be considered to have some overfitting, since the validation loss is higher than the training loss and it is possible that this gap may close by increasing the number of training samples even further (which may be seen as an example of poor generalizability on unseen data due to limited training data). Another extreme example is when we continue learning for longer epochs, the validation loss can decrease again, with double descent loss curves. This is usually not regarded as overfitting, but may fall under the overfitting category based on the explanation in the 1st paragraph of Section 1.
- In my opinion, trying to directly target overfitting is difficult since different researchers may have different definitions, so adopting a specific type of overfitting which "after a certain amount of training, the validation loss begins to increase while the training loss continues to decrease", seems like a good direction (although I wrote some issues above that perhaps should be discussed in the paper). However, if the paper adopts this direction, can we automatically label the loss history by checking if "after a certain amount of training, the validation loss begins to increase while the training loss continues to decrease", instead of using human labels, as explained in Step 3 of Section 4.1?
- It was also not clear to me how the two labellers decided the label of overfit/non-overfit.
- A related question is, how would we know that in papers (for constructing the real-world test dataset) with samples of overfitting, they are samples of the same type of overfitting?
- There are many papers that utilize the training history of maching learning models, but I am not sure if they were discussed. For example, "Learning Curves for Decision Making in Supervised Machine Learning - A Survey" discusses a lot of papers that use the training history or learning curves.
- For regression tasks, we usually use the same loss function for training and for evaluating, but for classification tasks, we may use the 01 loss function for evaluating the accuracy while using a different loss such as the cross entropy loss for training. This is known to cause interesting discrepancies between the behavior of the two different loss functions, e.g., discussed in Guo et al. "On Calibration of Modern Neural Networks" (ICML 2017) or Soudry et al. "The Implicit Bias of Gradient Descent on Separable Data" (JMLR 2018), and the increase of the validation loss may not be harmful in terms of the 01 loss function. Therefore, if we want to have better accuracy (i.e., lower 01 loss), is it possible that it may be better to use 01 loss instead of the cross entropy loss for validation in the proposed algorithm? (Perhaps this is not much of an issue, since the proposed method is helpful in achieving higher accuracy in the experiments, but wondering if this will make the proposed method even better.)

**Summary Of The Paper:**

The paper tries to detect and prevent overfitting by using the training loss and validation loss history. For detecting overfitting, the paper propose a strategy to collect a labelled dataset with binary labels: overfit and no overfit. The paper shows that a time-series classifier can be trained, and empirically showed to have high average F-scores, which is better than correlation based methods. The paper further shows a way to prevent overfitting, by using a detecting method on the whole observed history and by extracting the latest history with a rolling window. If we detect overfitting, we stop training and return the epoch that has the lowest validation loss. This seemed to work well, compared with an early stopping baseline.

**Summary Of The Review:**

The paper proposes an algorithm that detects and prevents overfitting, based on the labelled dataset that the paper constructed. This seems to be a new direction for research on detecting/preventing overfitting. However, discussions on how the labels were collected (or the two labellers' definition of overfitting) was not provided. A discussion about other papers that also use the training history was not discussed. Since the final target seems to be the accuracy in some of the experiments, using the validation 01 loss function seems to be a natural choice, rather than using the cross-entropy loss function. More discussions on this choice would be helpful. Overall, although the research direction is interesting, I decided to choose the score with 5.

---

> ### Author Response · Authors · 2022-11-19
> **First reply to reviewer xGGx (part 3/3)**
>
> ### Q4
>
> > A related question is, how would we know that in papers (for constructing the real-world test dataset) with samples of overfitting, they are samples of the same type of overfitting?
>
> Thank you for your question. We do not know whether the samples on the real-world test dataset are the same type of overfitting. What important isthat the labels (overfit vs. non-overfit) of the collected samples in the real-world dataset are reliable since we will use them as ground truth to evaluate our approach. To ensure the quality of the labeling, we only collect self-acknowledged (i.e., explicitly indicated by the authors of those papers) overfitting/non-overfitting training histories from top ML conferences and journals.
>
> -------
> ### Q5
>
> > There are many papers that utilize the training history of maching learning models, but I am not sure if they were discussed. For example, "Learning Curves for Decision Making in Supervised Machine Learning - A Survey" discusses a lot of papers that use the training history or learning curves.
>
> Thank you for your suggestion. We have added the discussion about the suggested paper in Section 1.
>
> -------
> ### Q6
>
> > For regression tasks, we usually use the same loss function for training and for evaluating, but for classification tasks, we may use the 01 loss function for evaluating the accuracy while using a different loss such as the cross entropy loss for training. This is known to cause interesting discrepancies between the behavior of the two different loss functions, e.g., discussed in Guo et al. "On Calibration of Modern Neural Networks" (ICML 2017) or Soudry et al. "The Implicit Bias of Gradient Descent on Separable Data" (JMLR 2018), and the increase of the validation loss may not be harmful in terms of the 01 loss function. Therefore, if we want to have better accuracy (i.e., lower 01 loss), is it possible that it may be better to use 01 loss instead of the cross entropy loss for validation in the proposed algorithm? (Perhaps this is not much of an issue, since the proposed method is helpful in achieving higher accuracy in the experiments, but wondering if this will make the proposed method even better.)
>
> Thank you so much for your suggestion. We have added the results of overfitting prevention methods based on zero-one loss (i.e., classification error) to Appendix G. The results show that using zero-one loss curves increases the accuracy (which is renamed to “optimal rate”, i.e., the percentage of times that the optimal epoch was achieved) of our approach with KNN-DTW (please see Figure 10), which still outperforms early stopping and other time series classifiers based on zero-one loss curves. However, the delay of the KNN-DTW also increases and stops later than early stopping when the window size is less than or equal to 40 epochs (please see Table 13 for details). Furthermore, we found that our approach with KNN-DTW has a higher optimal rate but longer delay while using the zero-one loss curves (compared to using the original loss curves).
>
> -------
> Thank you once again for your help in making this paper stronger. We hope our responses adequately addresses your concerns and we welcome your feedback.
>
> **Reference**
>
> - [1] Ying, Xue. "An overview of overfitting and its solutions." Journal of physics: Conference series. Vol. 1168. No. 2. IOP Publishing, 2019.
> - [2] Bejani, M.M., Ghatee, M. A systematic review on overfitting control in shallow and deep neural networks. Artif Intell Rev 54, 6391–6438 (2021).

---

> ### Author Response · Authors · 2022-11-19
> **First reply to reviewer xGGx (part 2/3)**
>
> ### Q2
>
> > In my opinion, trying to directly target overfitting is difficult since different researchers may have different definitions, so adopting a specific type of overfitting which "after a certain amount of training, the validation loss begins to increase while the training loss continues to decrease", seems like a good direction (although I wrote some issues above that perhaps should be discussed in the paper). However, if the paper adopts this direction, can we automatically label the loss history by checking if "after a certain amount of training, the validation loss begins to increase while the training loss continues to decrease", instead of using human labels, as explained in Step 3 of Section 4.1?
>
> Thank you for your advice. We have added Appendix H and updated Section 4.1 to include the results of using a heuristic method for labelling the training history automatically based on the following conditions:
>
> - The training loss and validation loss both decrease in the first $inc_p$ percentage of the training history.
> - The training loss and validation loss both decrease in the last $dec_p$ percentage of the training history.
> - The gap between the training loss and validation loss exceeds $gap_p$ percentage of the sum of the training and validation loss.
>
> We performed a grid search for selecting the best values for $inc_p$, $dec_p$, and $gap_p$. The best performance the heuristic method can achieve is a 0.75 F-score for the overfitting samples with 0.96 precision and 0.61 recall. The result shows that the heuristic method does not work well on the simulated dataset in comparison to other methods (see Table 7). Furthermore, the heuristic method performs poorly on the real-world dataset, with a 0.22 average F-score. Hence, human labels are still required for the time series classifier to automatically extract the rules for detecting overfitting whether or not there is overfitting.
>
> -------
> ### Q3
>
> > It was also not clear to me how the two labellers decided the label of overfit/non-overfit.
>
> Thank you for your comment. Two labellers independently labelled the samples based on the definition that we have updated in Section 1: “the training and validation losses of the overfit model both decrease at the beginning of the training process. Following that, the validation loss increases while the training loss decreases, resulting in a large gap between the training and validation losses”. After that, two labellers had two rounds of discussion to resolve conflicts for constructing the simulated dataset (see the details in Step 3 of Section 4.1).

---

> ### Author Response · Authors · 2022-11-19
> **First reply to reviewer xGGx (part 1/3)**
>
> ### Q1
>
> > If I understood the paper correctly, I feel the main issue is that the paper does not directly discuss what "overfitting" is. There are two sentences that try to address what overfitting is, in the abstract and introduction: 1) the abstract mentions that "overfitting of deep learning models on training data leads to poor generalizability on unseen data", and 2) 1st paragraph of Section 1 (Introduction) mentions that "for the overfit model, after a certain amount of training, the validation loss begins to increase while the training loss continues to decrease". The 1st one (in the abstract) can be regarded as a consequence for overfitting rather than an explanation of overfitting, so for example, the same sentence can be used for underfitting. The 2nd one (in the 1st paragraph of Section 1) is more specific, and if we use this as the definition of overfitting, it may exclude some learning curves which we may sometimes regard as overfitting or include ones that may not be overfitting. For example, if the validation loss stops decreasing after a certain number of epochs (but does not increase, just becomes flat) while the training loss keeps decreasing, the gap between training/validation loss will widen, and some people will regard this is a case of overfitting. As an extreme example, Fig 1 (b) can be considered to have some overfitting, since the validation loss is higher than the training loss and it is possible that this gap may close by increasing the number of training samples even further (which may be seen as an example of poor generalizability on unseen data due to limited training data). Another extreme example is when we continue learning for longer epochs, the validation loss can decrease again, with double descent loss curves. This is usually not regarded as overfitting, but may fall under the overfitting category based on the explanation in the 1st paragraph of Section 1.
>
> Thank you for your insightful comments and detailed examples. We agree that the two sentences in the abstract and introduction are insufficient to define overfitting and we have updated them in the manuscript. We have updated the first sentence in the abstract to “Overfitting occurs in deep learning models when instead of learning from the training data, they tend to memorize it, resulting in poor generalizability”, which adopts the definitions from prior work: “model does not generalize well from observed data to unseen data, which is called overfitting” [1] and “while with overfitting, the model is complex and only memorizes the training data with limited generalizability” [2]. This definition of overfitting is an abstract concept and we have updated the sentence you mentioned in Section 1 to make it more specific: “the training and validation losses of the overfit model both decrease at the beginning of the training process. Following that, the validation loss increases while the training loss decreases, resulting in a large gap between the training and validation losses”.
>
> However, people can still have different judgments about whether the three given examples are overfitting or not. As a result, developing a heuristic rule from manually specified criteria is challenging. Instead, in this paper, we decide to use time series classifiers to learn from human-labelled training histories in order to extract the common ideas of overfitting. Hence, the time series classifier can extract the rules for detecting overfitting automatically without requiring experts to develop the rules and set the corresponding threshold. Two labellers labelled the training histories in the simulated dataset independently and achieved 95% agreement (please see Step 3 of Section 4.1), implying that the two labellers share a common view of overfitting. To evaluate our approach, we collect self-acknowledged (i.e., explicitly indicated) overfitting/non-overfitting training histories from top ML conferences and journals (ensuring the quality of the labels) to construct the real-world dataset. Our approach can correctly identify overfitting samples on the simulated dataset (please see Table 7 in Appendix B) and achieve a 0.91 F-score on the real-world dataset (please see Table 1). These results show that the proposed time series classifier can extract the rules from the human-labelled training histories (the simulated dataset) and these rules can be generalized well to other datasets and models (the real-world dataset).

---

### Official Review · Reviewer_aLAk · 2022-10-18

**Confidence:** 4
**Correctness:** 3
**Technical Novelty And Significance:** 2
**Empirical Novelty And Significance:** 3
**Recommendation:** 5

**Clarity, Quality, Novelty And Reproducibility:**

The clarity is good. The quality is ok because the method is simple and does not have major flaw. The reproducibility is ok because the  replication package is provided.

**Strength And Weaknesses:**

Strengthes:
1. The paper is generally well written and easy to follow. The proposed method is also simple and intuitive.
2. The problem of detecting and preventing overfitting in deep learning is an important problem.
3. Experimental results show some advantages over prior methods for detecting and preventing overfitting in deep learning models.

Weaknesses:
1. The proposed method is straightforward and does not have much technical novelty/contribution. However, this is perfectly fine if the proposed method can be really useful and help facilitate training of deep learning models.
2. The experimental setup is not fully convincing. First, the test set contains only 40 data points, which is too small and thus vulnerable to randomness issues. Also, these papers are all collected in literature investigating the overfitting problem, which may results in similar choices of tasks/models and may not be able to reflect the trained classifier's real generalization ability on more realistic datasets and tasks. Also, the train histories explicitly labeled as overfitting or not and shown in the paper are probably typically overfitting examples that are very easy to distinguish. Moreover, the train set is manually labeled and require full training to generate, thus relatively laborsome.
3. The authors did not compare with intrusive methods because they require retraining, but their method also require training a classifier with labeled data. I do agree that these intrusive methods are not directly comparable with the proposed method, but it would be better that the authors at least report their performance in the experiments to help reader get a better understanding of the comparison.

**Summary Of The Paper:**

This paper proposes a new method to detect and prevent overfitting in training deep neural network models. The proposed method trains a time series classification model to detect overfitting based on the model's training history. The trained classifier can also be used to prevent overfitting by identifying the optimal point to stop a model's training. The authors manually labeled several hundred data points using existing datasets and test on several other data points collected from machine learning literature. Results show that the proposed approach outperforms conventional correlation-based methods on overfitting detection and slightly underperforms early-stopping for overfitting prevention but leads to lower delay, resulting in less wasted computation.

**Summary Of The Review:**

In sum, I think the paper investigate an interesting problem. However, the technical contribution is not strong enough and the empirical experiments does not fully support the claims. Therefore I would suggest the authors to revise and improve the empirical study in the future.

---

> ### Author Response · Authors · 2022-11-19
> **First reply to reviewer aLAk (part 2/2)**
>
> ### Q2.2
>
> > Also, the train histories explicitly labeled as overfitting or not and shown in the paper are probably typically overfitting examples that are very easy to distinguish.
>
> The collected training histories are all self-acknowledged samples which are mostly easy to distinguish by domain experts. However, it is not an easy task for detecting overfitting automatically. Table 1 shows that the best F-score of the baseline methods is 0.86, and our proposed method cannot perfectly distinguish them either (with a 0.91 F-score). We hope this paper may shed a light on this problem and inspire more research.
>
> -------
> ### Q2.3
> > Moreover, the train set is manually labeled and require full training to generate, thus relatively laborsome.
>
> We agree that generating training histories and manually labelling them is laborsome. That is why we share all of the training histories and labels in our [replication package](https://github.com/anonymous-p/overfit_detect/tree/main/data/training) so that other researchers can use our data and labels. Furthermore, other researchers can use the trained time series classifiers directly from our [replication package](https://github.com/anonymous-p/overfit_detect/tree/main/models). We have updated the text in Section 6 to clarify this point.
>
> -------
> ### Q3
>
> > The authors did not compare with intrusive methods because they require retraining, but their method also require training a classifier with labeled data. I do agree that these intrusive methods are not directly comparable with the proposed method, but it would be better that the authors at least report their performance in the experiments to help reader get a better understanding of the comparison.
>
> Thank you for your suggestion. We have added the results of using perturbed validation (PV) [1] to Appendix D. The PV method injects three levels of noise (0.1, 0.2, and 0.3) into the labels in the training set and retrains the model. The PV measurement will show how much the accuracy decreases in response to the injected noise and indicate if overfitting is present. The results show that the PV method takes a significant amount of computational time for retraining models but results in poor performance (i.e., 0.47 F-score) on the simulated dataset (please see Table 10 in Appendix D). One possible reason behind the poor performance could be that the PV method is designed for comparing different models for model selection and does not work well in the scenario with a single model structure. We do not calculate the PV measurement on the real-world dataset since retraining the models takes too long, and the performance on the simulated dataset does not justify this extra effort.
>
> -------
> Thank you once again for your help in making this paper stronger. We hope our response addresses your concerns and welcome your feedback.
>
> **Reference**
>
> - [1] J. Zhang, E. T. Barr, B. Guedj, M. Harman, and J. Shawe-Taylor, “Perturbed Model Validation: A New Framework to Validate Model Relevance,” p. 11.

---

> > ### Comment · Reviewer_aLAk · 2022-11-30
> > **Reply to the authors' response**
> >
> > Thank you for your responses. The updates partially address some of the concerns. However, I still believe the technically contribution is not very significant. (as pointed out by Reviewer xGGx, a lot of paper and researchers use training history to decide overfitting.) And since overfitting is kind of subjective, I'm not convinced that using a trained model can be helpful in general. The experiments are also not fully convincing because the settings are mostly artificial and kind of toy. Therefore I will keep the score as it is.

---

> ### Author Response · Authors · 2022-11-19
> **First reply to reviewer aLAk (part 1/2)**
>
> ### Q1
>
> > The proposed method is straightforward and does not have much technical novelty/contribution. However, this is perfectly fine if the proposed method can be really useful and help facilitate training of deep learning models.
>
> Thank you for your comment. As Reviewer AtDu said "this is the first paper I am aware of that uses time classification to detect and prevent overfitting" and we believe our method is novel. Furthermore, we agree with your comment that the problem of detecting and preventing overfitting in deep learning is an important problem. To address the problem, our method obtains a 0.91 F-score for overfitting detection (Table 1) and stops training earlier than early stopping with higher accuracy for overfitting prevention (Figure 5 and Tables 2 & 3). We agree that our method is simple and intuitive (similar to early stopping in that regard), however, we also consider this a strength of our method as it will be easy to use for researchers and practitioners (just like early stopping is).
>
> -------
> ### Q2.1
>
> > The experimental setup is not fully convincing. First, the test set contains only 40 data points, which is too small and thus vulnerable to randomness issues. Also, these papers are all collected in literature investigating the overfitting problem, which may results in similar choices of tasks/models and may not be able to reflect the trained classifier's real generalization ability on more realistic datasets and tasks.
>
> Thank you for your comments. To address your concern, we conducted a 3-fold cross validation on our simulated dataset (since collecting more real world examples of overfit is hard and expensive - more details are provided later in the response) and present the average F-score and its variance here. If the results are due to randomness, we might observe a low F-score and high variance and vice versa. Upon conducting 3 fold cross-validation, our studied time series classifiers (except HMM-GMM) achieve an F-score of more than 0.95 on the validation set in a 3-fold cross validation (we have included the results in Table 6 of Appendix B). These results show that our results are not due to randomness.
>
> Also, while the 40 data points in the real-world dataset are using similar datasets and models, such as VGG-16 on CIFAR-100 and ResNet-18 on Tiny-ImageNet, the datasets and models in the real-world dataset (for evaluation) are completely different from those in the simulated dataset (for training). In comparison to the real-world dataset, the datasets (please see Appendix A) and model architectures (i.e., NN with one or two hidden layers) in the simulated dataset are relatively simple. Table 1 shows that the trained time series classifier generalizes well from the simulated dataset to the real-world dataset (with a 0.91 F-score).
>
> Additional data points for the real-world dataset are difficult to obtain since we only collect self-acknowledged (i.e., explicitly indicated) overfitting/non-overfitting training histories from top ML conferences and journals to ensure the quality of the labelling. We need reliable labels to construct the dataset for evaluating our approach since there is no prior dataset of labelled training histories (overfit vs. non-overfit) or prior research about this problem. After surveying 17 conferences and 12 publications in the previous 5 years based on the CORE and CCF rating systems (please see details in Section 4.1), we could only find these 40 data points. These data points are representative of the examined literature and serve as a basic starting point for future research.

---

### Official Review · Reviewer_AtDu · 2022-10-23

**Confidence:** 4
**Correctness:** 3
**Technical Novelty And Significance:** 3
**Empirical Novelty And Significance:** 3
**Recommendation:** 6

**Clarity, Quality, Novelty And Reproducibility:**

Clarity: The paper is well written in modular sections that define and motivate the problem, other solutions in the literature, their proposed method, and a reasonable experiments and results section. It is easy to understand and flows well.
Quality: The work follows a standard scientific approach of problem definition, description of existing solutions, a proposal of a new method, experiments and results. The experiments section uses both simulated and real datasets. Collecting and reproducing the training and validation loss histories from 30+ papers is a good contribution
Novelty: While the problem is important, the idea is fairly obvious, and it is an incremental contribution
Reproducibility: The authors have published the code, datasets and results on github. It seems fairly easy to reproduce.

**Strength And Weaknesses:**

Strengths:
1. The paper uses a simple time series classifier to learn the regions of overfitting using training and validation losses. It is a non-destructive method and can be applied easily without any modification of the training algorithm itself.
2. The approach is tested well using a number of datasets from published models. The authors do a good job of collecting or reproducing the training and validation loss histories from 30+ papers. This is a useful contribution

Weaknesses:
1. From the paper, I struggled to infer they train a single model using all training and validation loss histories, or if they train a model per dataset. If the former, how well do they think this model generalizes to other datasets? i.e. is this a single pre-trained model that others can use off the shelf, or do other researchers need to train a model per dataset?
2. There is very little discussion of why some methods (like KNN-DTW) are able to do well, while others (like HMM-GMM and BOSSVS) perform so poorly (Table 2). As such, this feels like a brute force search method of different algorithms for time series classification. The proposed candidate KNN-DTW is an expensive algorithm to implement for inference (it is the slowest inference algorithm among the ones they tested in Table 2).

**Summary Of The Paper:**

Detecting overfitting of deep learning networks in an efficient, accurate and non-intrusive manner is a non-trivial problem. The authors propose a method to learn a classifier that uses the history of training and validation losses to identify overfitting regions. The paper evaluates the approach using a number of loss histories of published models. They show that classification is more accurate in detecting overfitting compared to correlation methods, and that one of the classifiers KNN-DTW is also better able to identify the optimal stopping point compared to early stopping method. While there are a number of approaches to detect and prevent overfitting, this is the first paper I am aware of that uses time classification to detect and prevent overfitting. The paper does a good job collecting or reproducing the histories of models from 30+ papers. This is a useful contribution for further research in this topic.

**Summary Of The Review:**

The paper addresses an important problem in training deep learning networks  - viz. when to stop training, and doing it in a non-intrusive manner. It proposes a fairly obvious idea of training time series classifiers on histories of training and validation losses to detect regions of overfitting, and compares it favorably against existing methods based on correlation and early stopping. A good contribution is that the paper has collected or reproduced the loss histories from a number of other published papers. This is a useful contribution.

---

> ### Author Response · Authors · 2022-11-19
> **First reply to reviewer AtDu (part 2/2)**
>
> ### Q2.1
>
> > There is very little discussion of why some methods (like KNN-DTW) are able to do well, while others (like HMM-GMM and BOSSVS) perform so poorly (Table 2). As such, this feels like a brute force search method of different algorithms for time series classification.
>
> To be honest, we are not entirely sure why these algorithms perform so differently for overfitting detection, but we provide our intuition behind why different time-series classifiers have different performance in a discussion section.
>
> One reason for the performance of KNN-DTW might be that DTW is good at measuring similarity across curves, which aids KNN in distinguishing between overfit and non-overfit samples. In contrast, HMM-GMM performs poorly on both the simulated training and real-world test datasets. One possible explanation is that the extracted state models (via HMM) of the curves do not follow a Gaussian probability distribution. BOSSVS may be overfitted to the simulated dataset with a 1.0 F-score (please see Appendix B), resulting in a 0.67 F-score on the real-world dataset (Table 1). We have added this discussion to Appendix B.
>
> We further wish to clarify that we did not choose these algorithms through a brute-force search. Since there has been no prior systematic research on using time series classifiers for training histories as a reference, we chose the classifiers that were reported as baselines or state-of-the-art in several prior studies [3, 4, 5, 6].
>
> -------
> ### Q2.2
>
> > The proposed candidate KNN-DTW is an expensive algorithm to implement for inference (it is the slowest inference algorithm among the ones they tested in Table 2)
>
> We agree that KNN-DTW is relatively more expensive than other time series classifiers in Table 1 but it is still helpful. On the one hand, the inference speed of KNN-DTW is not prohibitive in practice since overfitting detection is only executed once for inspecting a target model after the training is complete. On the other hand, the inference speed of KNN-DTW might be an issue for overfitting prevention if the target model iterates each epoch relatively fast (e.g., training a simple model) in the training process.
>
> To address the concern, we already used a faster version of DTW with a time complexity of O(n) [2], but using KNN with DTW is still computationally intensive. We encourage future work to optimize time series classifiers to enable real-time overfitting detection and prevention.
>
> -------
> We hope that our answers address the reviewer's concerns, and we are glad to provide more clarifications if the reviewer has any other questions. Thank you again for your review!
>
> **Reference**
>
> - [1] T. Chen, Z. Zhang, S. Liu, S. Chang, and Z. Wang, “Robust Overfitting may be mitigated by properly learned smoothening,” presented at the International Conference on Learning Representations, Feb. 2022.
> - [2] Stan Salvador, and Philip Chan. “FastDTW: Toward accurate dynamic time warping in linear time and space.” Intelligent Data Analysis 11.5 (2007): 561-580.
> - [3] Varol, O., Ferrara, E., Menczer, F., & Flammini, A. (2017). "Early detection of promoted campaigns on social media," EPJ data science 6 (2017): 1-19.
> - [4] X. Xi, E. Keogh, C. Shelton, L. Wei, and C. A. Ratanamahatana, “Fast time series classification using numerosity reduction,” in Proceedings of the 23rd international conference on Machine learning, New York, NY, USA, Jun. 2006, pp. 1033–1040. doi: 10.1145/1143844.1143974.
> - [5] Anami, B.S., Bhandage, V.A. A Comparative Study of Suitability of Certain Features in Classification of Bharatanatyam Mudra Images Using Artificial Neural Network. Neural Process Lett 50, 741–769 (2019).
> - [6] Z. Wang, L. Wang, C. Huang, Z. Zhang and X. Luo, "Soil-Moisture-Sensor-Based Automated Soil Water Content Cycle Classification With a Hybrid Symbolic Aggregate Approximation Algorithm," in IEEE Internet of Things Journal, vol. 8, no. 18, pp. 14003-14012, 15 Sept.15, 2021, doi: 10.1109/JIOT.2021.3068379.

---

> ### Author Response · Authors · 2022-11-19
> **First reply to reviewer AtDu (part 1/2)**
>
> ### Q1.1
>
> > From the paper, I struggled to infer they train a single model using all training and validation loss histories, or if they train a model per dataset.
>
> We apologize for the unclear description of the data we use for training the model (i.e., the time series classifier). We train a time series classifier (i.e., a single model) using all the training and validation loss histories from the simulated dataset (which contains 419 labelled training histories). We have updated the manuscript in Section 4 to make it clearer.
>
> -------
> ### Q1.2
>
> > If the former, how well do they think this model generalizes to other datasets? i.e. is this a single pre-trained model that others can use off the shelf, or do other researchers need to train a model per dataset?
>
> Thank you for the questions. It is a very important point of clarification we wish to make. Other researchers do not need to train their own models; instead, they can use our pre-trained model (i.e., our trained time series classifier) from our [replication package](https://github.com/anonymous-p/overfit_detect/tree/main/models). We have updated Section 6 of the manuscript to clarify this point.
>
> We show that the trained time series classifier generalizes well to other datasets since it achieves a 0.91 F-score (Table 1) on the real-world dataset (for evaluation), which contains more complex datasets and model architectures than the simulated dataset (for training). In particular, the datasets in the simulated dataset come from the UCI repository (please see Appendix A) and the model architectures are NNs with one or two hidden layers. In contrast, the real-world dataset includes training histories from complex models and datasets such as VGG-16 on CIFAR-100 and ResNet-18 on Tiny-ImageNet [1].

---

### Official Review · Reviewer_wxLr · 2022-10-24

**Confidence:** 4
**Correctness:** 4
**Technical Novelty And Significance:** 2
**Empirical Novelty And Significance:** 3
**Recommendation:** 6

**Clarity, Quality, Novelty And Reproducibility:**

The paper is very clear and easy to read, and there is a corresponding Github repository.

**Strength And Weaknesses:**

+ The proposed approach seems novel and is intuitively plausible
+ The results indicate that the method is preferable to existing correlation-based approaches to detecting overfitting
+ The paper is easy to follow and code is provided

- Accuracy is measured with respect to the distance to the "optimum" stopping point, not predictive accuracy of the neural network model

What matters is the predictive performance of the model, *not* how close we are to the "optimum" stopping point.


**Summary Of The Paper:**

The submission presents a method for improved identification of an early-stopping point based on validation loss curves. This method is based on training a time series classifier on validation loss curves that have been labeled as "overfit" vs. "not overfit", using a collection of UCI-type datasets and various network architectures. These training examples for the time series classifiers were labeled by the authors. Experiments compare various time series classifiers and find that a k-NN-based approach, with dynamic time warping, based on a rolling window, works best. This approach to detecting overfitting (and, thus, enabling early stopping) outperforms existing methods for overfitting detection based on measuring correlation between training and validation loss curves. Further experiments, comparing to plain early stopping using various values for the patience parameter in both the proposed method and standard early stopping, show that the proposed method more accurately identifies the stopping point with minimum validation loss than the standard plain patience-based approach.

**Summary Of The Review:**

The paper is interesting, and the proposed approach may have merit.

---

> ### Author Response · Authors · 2022-11-19
> **First reply to reviewer wxLr (part 2/2)**
>
> ### Q3
>
> > Early stopping is based on validation loss curves, not classification error (i.e., zero-one loss)
>
> &
>
> > To my knowledge, early stopping based on validation loss is generally not recommended when the goal is to maximize classification accuracy: classification accuracy may still increase when loss starts to increase.
>
> Thank you for your suggestion. We do not find research that recommends that early stopping based on validation loss should be avoided for classification tasks. In contrast, we notice that early stopping based on validation loss curves is frequently used by researchers for classification tasks [5, 6, 7, 8, 9].
>
> To address your concerns, we included Appendix G, which contains the results of early stopping and our approach based on zero-one validation loss. Compared to the early stopping based on validation loss (please see Figure 5), the optimal rate of early stopping based on zero-one loss is quite similar (please see Figure 10).
>
> The results show that the optimal rate improves (with an average of 2.5%) when the patience value is between 35 and 90 epochs but declines (with an average of 2.9%) when the patience value is less than 35 epochs. For our method with KNN-DTW, the results show that using zero-one loss curves increases the optimal rate, which still outperforms early stopping and other time series classifiers (please see Figure 10). However, the delay of the KNN-DTW also increases and stops later than early stopping when the window size is less than or equal to 40 epochs (please see Table 13 for details). Furthermore, we found that our approach with KNN-DTW has a higher optimal rate but longer delay while using the zero-one loss curves (compared to using the original loss curves).
>
>
> -------
> ### Q4
>
> > No significance testing is performed to establish that observed differences are statistically significant
>
> Thank you for your suggestion. We have added significance testing in Appendix F and updated Section 5 for the delay between early stopping and our approach. The results show that the delay of our method with KNN-DTW is significantly different from early stopping and the effect size is medium when the patience/window size is set to 40 epochs. The effect size becomes large when the patience/window size is greater than 40 epochs.
>
> -------
> ### Q5
> > "outperforms the state-of-the-art by at least 5%" - 5% of what?
>
> Thank you for the question. We mean 5% of the F-score (as shown in Table 1) compared to the next best-performing overfitting detection method. For instance, our method with KNN-DTW achieves a 0.91 F-score whereas the correlation-based method with Spearman metric (the best performance in baselines) achieves a 0.86 F-score. We have updated the manuscript in Section 1 to make it clearer.
>
> -------
> Thank you again for your review! We hope our response addresses your concerns and look forward to hearing from you.
>
> **Reference**
>
>
> - [1] R. H. Shumway and D. S. Stoffer, "Chapter 2.3 - Smoothing in the Time Series Context" in Time Series Analysis and Its Applications: With R Examples. Cham: Springer International Publishing, 2017. doi: 10.1007/978-3-319-52452-8.
> - [2] K. Molugaram and G. S. Rao, “Chapter 12 - Analysis of Time Series,” in Statistical Techniques for Transportation Engineering, K. Molugaram and G. S. Rao, Eds. Butterworth-Heinemann, 2017, pp. 463–489. doi: 10.1016/B978-0-12-811555-8.00012-X.
> - [3] C. Guo, G. Pleiss, Y. Sun, and K. Q. Weinberger, “On Calibration of Modern Neural Networks,” in Proceedings of the 34th International Conference on Machine Learning, Jul. 2017, pp. 1321–1330.
> - [4] D. Soudry, E. Hoffer, M. S. Nacson, S. Gunasekar, and N. Srebro, “The Implicit Bias of Gradient Descent on Separable Data,” Journal of Machine Learning Research, vol. 19, no. 70, pp. 1–57, 2018.
> - [5] Z. Tayeb et al., “Validating Deep Neural Networks for Online Decoding of Motor Imagery Movements from EEG Signals,” Sensors, vol. 19, no. 1, Art. no. 1, Jan. 2019, doi: 10.3390/s19010210.
> - [6] Ü. Atila, M. Uçar, K. Akyol, and E. Uçar, “Plant leaf disease classification using EfficientNet deep learning model,” Ecological Informatics, vol. 61, p. 101182, Mar. 2021, doi: 10.1016/j.ecoinf.2020.101182.
> - [7] M. M. Badža and M. Č. Barjaktarović, “Classification of Brain Tumors from MRI Images Using a Convolutional Neural Network,” Applied Sciences, vol. 10, no. 6, Art. no. 6, Jan. 2020, doi: 10.3390/app10061999.
> - [8] B. Huang and K. M. Carley, “Parameterized Convolutional Neural Networks for Aspect Level Sentiment Classification.” arXiv, Sep. 13, 2019. doi: 10.48550/arXiv.1909.06276.
> - [9] F. Errica, M. Podda, D. Bacciu, and A. Micheli, “A Fair Comparison of Graph Neural Networks for Graph Classification.” arXiv, Feb. 17, 2022. doi: 10.48550/arXiv.1912.09893.

---

> ### Author Response · Authors · 2022-11-19
> **First reply to reviewer wxLr (part 1/2)**
>
> ### Q1
>
> > The most basic form of early stopping is used as the competitor in the experiments.
>
> &
>
> > One obvious competitor that should be included in the experiments, particularly considering the examples in Figure 6, is early stopping based on a smoothed loss curve.
>
> Thank you for your advice. We have added Appendix E to include the results of early stopping with the moving average [1, 2] to smooth the validation loss curve. However, it does not have better performance than the basic form of early stopping and cannot compete with our approach.
>
> For the example in Figure 6 (b), early stopping can achieve the optimal epoch (i.e., the epoch that leads to the optimal performance of our model on the validation set) after using the smoothed curve (please see Figure 8 in Appendix E).
>
> However, Table 11 shows that using a smoothed validation loss curve may hurt the accuracy (which is renamed to “optimal rate” according to Q2, i.e., the percentage of times that the optimal epoch was achieved in our experiments) and increase the delay. For example, the optimal rate drops from 48% to 38% when early stopping uses a smoothing window of 5 epochs and a patience parameter of 20 epochs.
>
> For better understanding, we have included an example of how early stopping cannot achieve the optimal stopping point after using the smoothed curve in Figure 9 of Appendix E. Furthermore, using a smoothed validation loss curve may increase the delay without improving the optimal rate. For instance, when setting the patience and smoothing window as 40 and 10 epochs respectively, the delay increases from 40 to 43.5 epochs but the optimal rate remains unchanged.
>
> -------
> ### Q2
>
> > Accuracy is measured with respect to the distance to the "optimum" stopping point, not predictive accuracy of the neural network model.
>
> &
>
> > What matters is the predictive performance of the model, not how close we are to the "optimum" stopping point.
>
> We apologize for the confusion. The optimal stopping point is defined as the epoch that has the optimal predictive performance for the model on the validation set, whereas the accuracy is defined as the percentage of times that the optimal stopping point is achieved (please see Section 4.3). For clarity, we changed “optimal stopping point” to “optimal epoch” and “accuracy” to “optimal rate”.
>
> **Why do we introduce the delay metric to measure the epoch difference between the stopped epoch and the best epoch?** For overfitting prevention methods, we are interested in not only the optimal rate but also how fast a method can decide to stop the training process (for saving time). For instance, if two overfitting prevention methods have the same optimal rate, we prefer to use the one which chooses the optimal epoch faster. The delay between the stopped epoch and the best epoch (which might not be the optimal epoch if the method stops too early) is the time required for overfitting prevention methods to make the decision. We use optimal rate and delay to evaluate the performance of overfitting prevention methods, similar to how we evaluate the accuracy and speed of an algorithm.

---

> > ### Comment · Reviewer_wxLr · 2022-11-24
> > **Acknowledgement of reply to reviewer wxLr**
> >
> > Thank you for your responses. I have increased my score to 6. I do think that the paper could be improved further by presenting results for the average classification accuracy of the models obtained using the various stopping methods. The question is whether this difference in classification accuracy is significant.

---

> > > ### Author Response · Authors · 2022-11-30
> > > **Second reply to reviewer wxLr (part 1/1)**
> > >
> > > Thank you for your swift reply. We are glad that your concerns were addressed. We appreciate the insightful feedback and positive assessment of our work.
> > >
> > > We performed statistical tests on the classification accuracy of the models (called accuracy for short) obtained using overfitting prevention methods. The accuracy of our method with KNN-DTW and early stopping is significantly different when using a patience/window size of 20 epochs, with KNN-DTW having 2.7% higher average accuracy than early stopping. However, the accuracy of KNN-DTW and early stopping is not significantly different and the average accuracy is very similar when using a large value for patience/window size (e.g., 40 epochs). The reason for the similar accuracy is that both KNN-DTW and early stopping reach an 80% optimal rate when using a window size of 40 epochs (as shown in Figures 11 and 5), indicating that the majority of the obtained classification accuracy is the same. We will add an Appendix I to the final paper (during the camera-ready process) to include the results (Tables 14 and 15) of comparing the average classification accuracy of the models obtained using our approach and early stopping.

---

### Author Response · Authors · 2022-11-19
**To all ICLR reviewers: Summary of changes**

We sincerely thank the four reviewers for the insightful feedback and comments. These have greatly helped us to improve the quality of our manuscript.

All the changes are highlighted in blue in the updated manuscript. The main changes are summarized as follows:

- We included an experiment for early stopping based on a smoothed loss curve per Reviewer wxLr’s suggestion (Appendix E).
- We included an experiment for early stopping and our approach based on zero-one validation loss curves per Reviewer wxLr’s and Reviewer xGGx’s suggestions (Appendix G).
- We added an experiment using perturbed validation (which is an intrusive method for overfitting detection) for comparison per Reviewer aLAk’s suggestion (Appendix D).
- We added an experiment using a heuristic method for labeling training histories automatically per Reviewer xGGx’s suggestion (Appendix H).
- We added the results of 3-fold cross-validation of our approach per Reviewer aLAk’s suggestion (Appendix B).
- We added significance testing for the delay between early stopping and our approach per Reviewer wxLr’s suggestion (Section 5, Appendix F)
- We added discussion about the performance of the time series classifiers and explained why we select these algorithms per Reviewer AtDu’s suggestion (Appendix B).
- We updated the definition of overfitting for clarification per Reviewer xGGx’s suggestion (Abstract, Section 1).

---

### Decision · Program_Chairs · 2023-01-20

**Decision:**

Reject

**Justification For Why Not Higher Score:**

Missing details and very limited empirical evaluation.

**Justification For Why Not Lower Score:**

N/A

**Metareview: Summary, Strengths And Weaknesses:**

The paper proposes to train time series classifiers on validation loss curves which have been labeled as "overfit" vs. "not overfit" to detect and prevent overfitting. The authors showed that their method is more accurate in detecting overfitting compared to correlation methods, and is also better at identifying the optimal stopping point compared to early stopping method on a collection of UCI datasets. Reviewer recommendations are mixed. While all the reviewers consider the proposal of learning time series classifier to detect overfitting interesting and plausible, they raise different concerns on the work, e.g., small scale of the evaluation datasets (40 data points) making the results susceptible to randomness, additional requirement to manually label validation loss curves as overfitting and non-overfitting. IMO, the paper left many questions unanswered: 1) Beyond finding exactly the optimal epoch to early stop, the generalization gap at the early stopping point detected by different methods should also be compared; 2)  It suggests that there are patterns in the loss validation curves, more than looking at the smoothed validation loss curve, that can help detect overfitting, but the paper offers neither insights nor analyses to shed light on what those pattern could be; 3) how sensitive is the method to noise in the "overfit" vs "not overfit" labels; 4) which time series classifier is used? Are the results sensitive to the specific choices of classifiers?